# Adjusting Initial Noise to Mitigate Memorization in Text-to-Image Diffusion Models

**Hyeonggeun Han**[1,2*]  **Sehwan Kim**[1*]  **Hyungjun Joo**[1,2]
**Sangwoo Hong**[3†]  **Jungwoo Lee**[1,2,4†]
[1]ECE & [2]NextQuantum, Seoul National University
[3]CSE, Konkuk University
[4]Hodoo AI Labs
{hygnhan, sehwankim, joohj911, junglee}@snu.ac.kr
swhong06@konkuk.ac.kr

## Abstract

Despite their impressive generative capabilities, text-to-image diffusion models often memorize and replicate training data, prompting serious concerns over privacy and copyright. Recent work has attributed this memorization to an attraction basin—a region where applying classifier-free guidance (CFG) steers the denoising trajectory toward memorized outputs—and has proposed deferring CFG application until the denoising trajectory escapes this basin. However, such delays often result in non-memorized images that are poorly aligned with the input prompts, highlighting the need to promote earlier escape so that CFG can be applied sooner in the denoising process. In this work, we show that the initial noise sample plays a crucial role in determining when this escape occurs. We empirically observe that different initial samples lead to varying escape times. Building on this insight, we propose two mitigation strategies that adjust the initial noise—either collectively or individually—to find and utilize initial samples that encourage earlier basin escape. These approaches significantly reduce memorization while preserving image-text alignment. Code is available at https://github.com/hygnhan/init_noise_diffusion_memorization.

## 1 Introduction

Text-to-image diffusion models have garnered significant attention for their remarkable ability to generate high-quality images that are semantically aligned with textual prompts. Despite their success, a growing body of work has revealed that these models are prone to memorizing and reproducing content from their training data, sometimes generating images that are nearly identical to those seen during training [39, 5, 40]. This memorization behavior poses serious concerns, including potential leakage of privacy-sensitive or copyrighted material [4, 23]. In addition to the legal and ethical risks, such behavior undermines the utility of diffusion models by limiting their generative diversity and reducing their capacity to synthesize truly novel content. These concerns highlight the pressing need to understand and mitigate memorization in diffusion models to ensure their safe and effective deployment.

A range of approaches have been proposed to mitigate memorization in diffusion models [49, 19, 34, 40]. Among them, inference-time mitigation strategies have gained particular attention due to their computational efficiency and practicality for real-world deployment. Several studies have linked

---

*These authors contributed equally to this work

†Corresponding authors

39th Conference on Neural Information Processing Systems (NeurIPS 2025).

memorization to specific prompt-image associations, motivating mitigation methods that perturb prompts during inference [40, 49]. Other researches have focused on internal model behaviors, particularly within cross-attention layers, revealing that memorization can be attributed to sharply focused attention maps [34] or a small set of specific neurons [19]. Building on these insights, mitigation techniques such as attention logit rescaling [34] and neuron deactivation [19] have been introduced.

A recent line of work has identified classifier-free guidance (CFG) [20] in the early stages of denoising as a key factor contributing to memorization in diffusion models [22]. To explain this phenomenon, Jain et al. [22] introduced the concept of an *attraction basin*—a region in the sample-time space where applying CFG consistently drives the generation process toward memorized outputs, regardless of the random initialization. To mitigate this effect, the authors proposed deferring the use of CFG until the denoising trajectory exits the attraction basin. However, they also highlighted a trade-off: applying CFG too late can lead to non-memorized images that are poorly aligned with the conditioning prompt. To address this issue, they emphasized the need to escape the attraction basin earlier in the generation process. Toward that end, the authors introduced opposite guidance, a strategy that applies inverted conditional signals during early denoising steps to push the trajectory away from the attraction basin, thereby enabling earlier and safer application of CFG.

In this paper, we propose a novel approach to mitigating memorization by enabling an earlier escape from the attraction basin. In contrast to prior efforts that modify the denoising trajectory through adjustments to guidance signals, we instead focus on the starting point of the trajectory. Our empirical findings reveal that different initial Gaussian noise samples result in significantly different escape times from the attraction basin. This observation motivates the hypothesis that certain initializations naturally lie closer to the basin's boundary, allowing the generation process to evade the memorization-inducing steering force more quickly. Building on this insight, we propose two inference-time mitigation strategies that find and utilize favorable initializations to promote early basin escape. Specifically, we propose a batch-wise method, which adjusts initial samples collectively, and a per-sample method, which adapts each initialization individually. Our approach offers a novel perspective that diverges from the conventional wisdom [49, 22] that the choice of initialization has minimal impact on memorization mitigation.

The key contributions of this paper are summarized as follows:

- We offer a novel perspective on the role of the initial noise sample in memorization mitigation, employing the concept of the attraction basin—a region in which classifier-free guidance (CFG) steers the denoising trajectory toward memorized outputs. We show that different initializations lead to varying escape times from this basin.

- We propose two inference-time mitigation strategies—Batch-wise and Per-sample approaches—that find and leverage initial samples likely to escape the attraction basin earlier. These methods adjust the initial samples to produce non-memorized images while promoting better alignment with the conditioning prompt.

- Our empirical evaluations demonstrate that the proposed initial sample adjustment strategies produce more text-aligned, non-memorized outputs than those without adjustment, and outperform existing inference-time mitigation baselines in reducing memorization.

## 2 Background

### 2.1 Diffusion models

Diffusion models [21, 42] aim to generate samples from a target data distribution $q(x)$. During training, a data sample $x_0 \sim q(x)$ is progressively corrupted by Gaussian noise over $T$ timesteps such that $x_T \sim \mathcal{N}(\mathbf{0}, \mathbf{I})$. At each timestep $t$, the forward noising process is defined as:

$$q(x_t|x_{t-1}) = \mathcal{N}(x_t; \sqrt{1 - \beta_t}x_{t-1}, \beta_t\mathbf{I}), \tag{1}$$

where $\beta_t$ denotes the variance schedule. The noise predictor $\epsilon_\theta$ is trained to predict the noise added at each timestep. At inference time, the sampling process begins from $x_T \sim \mathcal{N}(\mathbf{0}, \mathbf{I})$, and a sample is generated by iteratively denoising via the reverse process:

$$x_{t-1} = \frac{1}{\sqrt{\alpha_t}}\Big(x_t - \frac{1 - \alpha_t}{\sqrt{1 - \bar{\alpha}_t}}\epsilon_\theta(x_t, t)\Big), \tag{2}$$

where $\alpha_t = 1 - \beta_t$ and $\bar{\alpha}_t = \prod_{s=1}^{t} \alpha_s$.

Text-to-image diffusion models such as Stable Diffusion [35] usually utilize CFG to steer the generation process toward a desired conditioning input $y$. Given a null condition $y_{\text{null}}$ representing the unconditional case, CFG modifies the noise prediction at each timestep as:

$$\epsilon^{\text{CFG}}(x_t, t, y) = \epsilon_\theta(x_t, t, y_{\text{null}}) + w_{\text{CFG}}(\epsilon_\theta(x_t, t, y) - \epsilon_\theta(x_t, t, y_{\text{null}})), \quad (3)$$

where $w_{\text{CFG}}$ controls the strength of the guidance. For notational convenience, we denote the conditional noise prediction $\epsilon_\theta(x_t, t, y) - \epsilon_\theta(x_t, t, y_{\text{null}})$ as $\tilde{\epsilon}_\theta(x_t, t, y)$. It is worth noting that the trained noise predictor $\epsilon_\theta$ approximates the score function at each timestep [20, 43]: $\epsilon_\theta(x_t, t, y) \approx -\sqrt{1 - \bar{\alpha}_t}\nabla_{x_t} \log p_\theta(x_t|y)$ and $\epsilon_\theta(x_t, t, y_{\text{null}}) \approx -\sqrt{1 - \bar{\alpha}_t}\nabla_{x_t} \log p_\theta(x_t)$.

## 2.2 Attraction basin

The attraction basin refers to a region formed in the denoising trajectory where applying CFG exerts a strong steering force toward memorized outputs. To mitigate memorization, Jain et al. [22] proposed deferring the application of CFG until the trajectory escapes this basin. A key aspect of this strategy is identifying when this escape occurs. They observed that when sampling proceeds without CFG, the magnitude of the conditional noise prediction $||\tilde{\epsilon}_\theta(x_t, t, y)||_2$ remains high within the basin. This is interpreted as empirical evidence of the memorization-inducing force present in this region. Just before the escape, the magnitude drops sharply—this timestep is referred to as the transition point. This behavior is illustrated in Figure 1. Applying CFG before the transition point (*i.e.*, within the attraction basin) typically results in memorized outputs, whereas applying it after the transition point leads to non-memorized images. However, excessively delaying CFG to avoid memorization may result in poor alignment with the

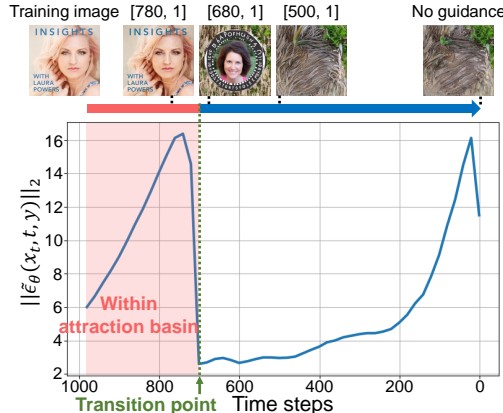

Figure 1: The magnitude of $\tilde{\epsilon}_\theta(x_t, t, y)$ at each timestep during sampling without CFG for a memorized prompt. Each image above corresponds to an output generated when CFG is applied over the timestep interval indicated by the associated square brackets.

conditioning prompt. This trade-off underscores the importance of encouraging earlier escape from the basin—*i.e.*, shifting the transition point to an earlier timestep—to generate non-memorized images that remain faithful to the prompt.

# 3 Initial sample adjustment for mitigating memorization

## 3.1 Motivation

To effectively mitigate memorization, CFG should be applied after the denoising trajectory escapes the attraction basin—*i.e.*, after the transition point. This motivates the need to shift the transition point to an earlier timestep. To this end, we investigate the influence of the initial Gaussian sample $x_T$, which defines the starting point of the trajectory. As shown in Figure 2, we consistently observe across different memorized prompts that varying $x_T$ results in different transition points. This suggests that the location of $x_T$, relative to the attraction basin, influences how quickly the trajectory escapes. Based on this observation, we conjecture that initial samples closer to the basin's boundary facilitate earlier escapes and thus earlier transition points. To validate this conjecture, we require a proxy for estimating a sample's proximity to the basin's boundary. Prior work [22] reports that the magnitude of the conditional noise prediction, $||\tilde{\epsilon}_\theta(x_t, t, y)||_2$, remains elevated within the basin and drops sharply upon exit. Leveraging this insight, we hypothesize that initial samples with smaller magnitudes $||\tilde{\epsilon}_\theta(x_T, T, y)||_2$ are more likely to escape the basin earlier. To test this hypothesis, we introduce a novel method that adjusts $x_T$ to obtain a modified sample $\tilde{x}_T$ with reduced magnitude of conditional guidance $||\tilde{\epsilon}_\theta(\tilde{x}_T, T, y)||_2$. We then analyze the magnitude of the conditional noise predictions throughout the denoising process.

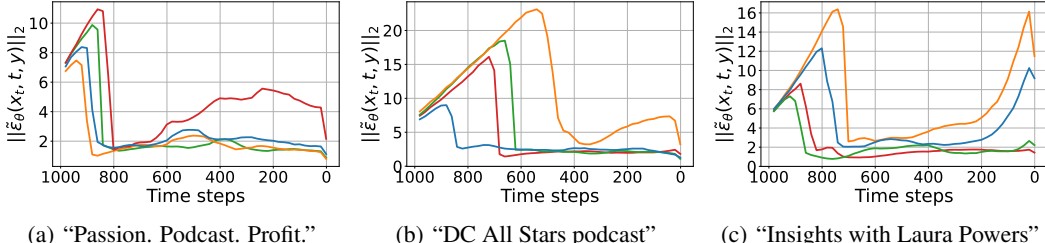

(a) "Passion. Podcast. Profit."     (b) "DC All Stars podcast"     (c) "Insights with Laura Powers"

Figure 2: The magnitude of the conditional noise prediction at each timestep during sampling without CFG for three memorized prompts. Each line color corresponds to a different initial Gaussian sample. Transition points occur at different timesteps depending on the choice of initial sample.

### 3.1.1 Initial sample adjustment using sharpness

To facilitate clear analysis of the impact of $||\tilde{\epsilon}_\theta(x_T, T, y)||_2$, we introduce a one-step adjustment method that modifies the initial sample $x_T$ using a tunable step size that controls the strength of the magnitude reduction. To this end, we adopt the sharpness definition from Foret et al. [16] to reduce the magnitude of the conditional noise prediction. According to prior works [20, 11], the conditional noise prediction can be approximated in terms of the score function:

$$\tilde{\epsilon}_\theta(x_t, t, y) \approx -\sqrt{1 - \bar{\alpha}_t} \nabla_{x_t} \log p_\theta(y|x_t). \tag{4}$$

Letting $L(x_t) = -\log p_\theta(y|x_t)$, we obtain $||\tilde{\epsilon}_\theta(x_t, t, y)||_2 \approx \sqrt{1 - \bar{\alpha}_t}||\nabla_{x_t} L(x_t)||_2$, indicating that reducing the magnitude of conditional guidance corresponds to minimizing the gradient norm of $L(x_t)$. To achieve this, we adopt the following sharpness measure:

$$L_{\text{sharp}}(x_t) = \max_{||\delta||_2 \leq \rho} L(x_t + \delta) - L(x_t). \tag{5}$$

This definition captures the worst-case local increase in $L$ within an $\ell_2$-ball of radius $\rho$ centered at $x_t$, and penalizing this sharpness term has been shown to correspond to gradient norm minimization [16, 53]. Based on this formulation, we obtain the adjusted initial sample $\tilde{x}_T$, which yields the reduced magnitude of conditional guidance, as follows:

$$\tilde{x}_T = x_T - \gamma \nabla_{x_T} L_{\text{sharp}}(x_T), \tag{6}$$

where $\gamma$ is a step size hyperparameter. To compute $\nabla_{x_T} L_{\text{sharp}}(x_T)$, we follow Foret et al. [16] and first approximate the inner maximization in Equation (5) using a first-order Taylor expansion to obtain

$$\delta^*(x_T) = \underset{||\delta||_2 \leq \rho}{\operatorname{argmax}} L(x_T + \delta) \approx \underset{||\delta||_2 \leq \rho}{\operatorname{argmax}} L(x_T) + \delta^\top \nabla_{x_T} L(x_T) = \underset{||\delta||_2 \leq \rho}{\operatorname{argmax}} \delta^\top \nabla_{x_T} L(x_T). \tag{7}$$

Since this approximation can be seen as a classical dual norm problem, its solution $\hat{\delta}(x_T)$ is given by:

$$\hat{\delta}(x_T) = \rho \cdot \frac{\nabla_{x_T} L(x_T)}{||\nabla_{x_T} L(x_T)||_2}. \tag{8}$$

By substituting $\hat{\delta}(x_T)$ from Equation (8) into the sharpness definition in Equation (5), and differentiating with respect to $x_T$, we obtain the update rule in Equation (6) as follows:

$$\begin{aligned} \tilde{x}_T &= x_T - \gamma \nabla_{x_T} (L(x_T + \hat{\delta}(x_T)) - L(x_T)) \\ &\approx x_T - \gamma (\nabla_{x_T} L(x_T)|_{x_T + \hat{\delta}(x_T)} - \nabla_{x_T} L(x_T)). \end{aligned} \tag{9}$$

Here, we follow Foret et al. [16] and approximate the gradient by omitting second-order terms. Substituting $\nabla_{x_T} L(x_T) \approx \frac{1}{\sqrt{1-\bar{\alpha}_T}} \tilde{\epsilon}_\theta(x_T, T, y)$, we express the final adjustment rule in terms of the noise predictor:

$$\tilde{x}_T \approx x_T - \tilde{\gamma}(\tilde{\epsilon}_\theta(x_T + \hat{\delta}(x_T), T, y) - \tilde{\epsilon}_\theta(x_T, T, y)), \tag{10}$$

where $\hat{\delta}(x_T) = \rho \cdot \frac{\tilde{\epsilon}_\theta(x_T, T, y)}{||\tilde{\epsilon}_\theta(x_T, T, y)||_2}$ and $\tilde{\gamma} = \frac{\gamma}{\sqrt{1-\bar{\alpha}_T}}$. Both $\tilde{\gamma}$ and $\rho$ are treated as hyperparameters.

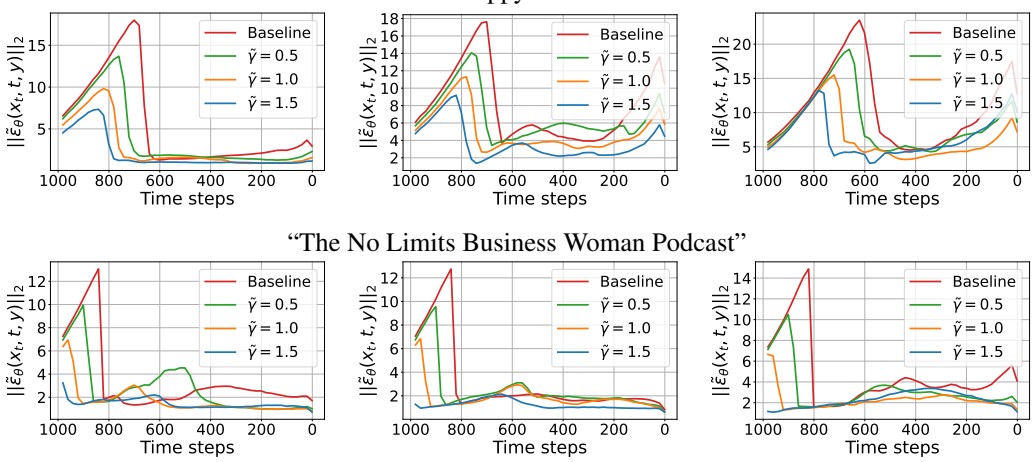

Figure 3: Plots showing the magnitude of the conditional noise prediction at each timestep during sampling without CFG under different adjustment strengths $\tilde{\gamma}$ for two memorized prompts. Each row corresponds to a different memorized prompt, and each column corresponds to a different initial Gaussian sample $x_T$. The baseline corresponds to $\tilde{\gamma} = 0$.

### 3.1.2 Magnitude analysis of conditional guidance for adjusted initial samples

In order to test our hypothesis, we use the adjusted initial sample $\tilde{x}_T$ obtained via Equation (10) and observe the $\ell_2$-norm of the conditional noise prediction at each timestep when CFG is not applied. We employ Stable Diffusion [35] with $\rho = 50$ and vary the adjustment strength $\tilde{\gamma} \in \{0, 0.5, 1.0, 1.5\}$ to investigate the influence of the magnitude of conditional noise prediction at the initial timestep $T$. Figure 3 presents the results across different memorized prompts and initial samples. Each row corresponds to a different prompt, while each column shows the effect of applying the proposed adjustment to a different initial sample. The baseline denotes the unadjusted case ($\tilde{\gamma} = 0$). First, we observe that the magnitude of the conditional noise prediction at the initial timestep $T$ decreases as $\tilde{\gamma}$ increases. This confirms that the proposed adjustment method successfully adjusts $x_T$ to yield an updated sample $\tilde{x}_T$ with reduced magnitude $||\tilde{\epsilon}_\theta(\tilde{x}_T, T, y)||_2$, as intended. Second—and more notably—we find that the transition point consistently occurs earlier in the denoising process as $\tilde{\gamma}$ increases. This trend holds across different prompts and initial samples, although the exact degree of the shift varies. These results support our hypothesis that initial samples with smaller magnitudes of conditional guidance $||\tilde{\epsilon}_\theta(x_T, T, y)||_2$ are more likely to escape the attraction basin earlier, thereby advancing the transition point.

## 3.2 Proposed mitigation methods

Motivated by the above observations, we propose two mitigation approaches: Batch-wise mitigation and Per-sample mitigation. Batch-wise mitigation adjusts initial samples $x_T$ collectively across a batch. This method reduces the magnitude of the conditional noise prediction using a small number of additional function evaluations (NFEs), producing adjusted samples $\tilde{x}_T$ with minimal computational overhead. In contrast, Per-sample mitigation modifies each sample $x_T$ individually. It reduces the magnitude of conditional noise prediction through direct backpropagation, continuing the update until the transition point is eliminated for each generation process. As a result, this method does not require an additional hyperparameter—a predefined timestep at which to begin applying CFG.

### 3.2.1 Batch-wise mitigation

For Batch-wise mitigation, we adopt the adjustment strategy introduced in the previous section. Given a batch of initial samples $x_T$, we apply the update rule in Equation (10) to the entire batch simultaneously. This update is repeated $M$ times, where $M$ is a tunable hyperparameter that determines the number of adjustment steps. The resulting adjusted samples $\tilde{x}_T$ are then used as initial samples for the image generation process. Additionally, following Jain et al. [22], we apply

**Algorithm 1:** Batch-wise mitigation

---

**Input:** adjustment strength $\tilde{\gamma}$, sharpness parameter $\rho$, number of adjustments $M$, CFG application start timestep $\tau$

**Output:** generated images $x_0$

Draw a batch of initial noise samples $x_T \sim \mathcal{N}(\mathbf{0}, \mathbf{I})$

/* Adjust the initial samples $x_T$ with a small number of additional NFEs */

**for** $i = 1$ **to** $M$ **do**

$\quad \hat{\delta}(x_T) = \rho \cdot \frac{\tilde{\epsilon}_\theta(x_T, T, y)}{||\tilde{\epsilon}_\theta(x_T, T, y)||_2}$

$\quad \tilde{x}_T = x_T - \tilde{\gamma}\big(\tilde{\epsilon}_\theta(x_T + \hat{\delta}(x_T), T, y) - \tilde{\epsilon}_\theta(x_T, T, y)\big)$

$\quad x_T = \tilde{x}_T$

**end**

/* Perform the denoising process using the adjusted initial samples */

**for** $t = T$ **to** $1$ **do**

$\quad$ **if** $t > \tau$ **then**

$\quad\quad \epsilon^{\text{CFG}}(x_t, t, y) = \epsilon_\theta(x_t, t, y_{\text{null}})$

$\quad$ **else**

$\quad\quad \epsilon^{\text{CFG}}(x_t, t, y) = \epsilon_\theta(x_t, t, y_{\text{null}}) + w_{\text{CFG}}(\epsilon_\theta(x_t, t, y) - \epsilon_\theta(x_t, t, y_{\text{null}}))$

$\quad$ **end**

$\quad x_{t-1} = \frac{1}{\sqrt{\alpha_t}}\Big(x_t - \frac{1 - \alpha_t}{\sqrt{1 - \bar{\alpha}_t}}\epsilon^{\text{CFG}}(x_t, t, y)\Big)$

**end**

---

CFG after a certain timestep to avoid generating memorized outputs. The complete procedure for Batch-wise mitigation is outlined in Algorithm 1.

### 3.2.2 Per-sample mitigation

For Per-sample mitigation, we adjust each initial sample $x_T$ individually by directly minimizing $||\tilde{\epsilon}_\theta(x_T, T, y)||_2$ via backpropagation. The adjustment continues until $||\tilde{\epsilon}_\theta(x_T, T, y)||_2$ falls below a predefined target loss $l_{\text{target}}$. In contrast to Batch-wise mitigation, CFG is applied from the beginning of the sampling process. As shown in the rightmost plot of the bottom row in Figure 3, when the magnitude is sufficiently minimized (*e.g.*, $\tilde{\gamma} = 1.5$), the transition point effectively disappears. This suggests that early application of CFG no longer induces memorization, thereby eliminating the need to predefine a specific timestep at which to begin applying CFG. Consequently, Per-sample mitigation enables the generation of non-memorized images even when CFG is applied from the initial timestep $T$. We conjecture that this behavior arises because sufficiently strong minimization yields an initial sample that lies outside the attraction basin. The pseudocode for Per-sample mitigation is provided in Appendix A.

## 4 Experiments

### 4.1 Experimental setup

**Diffusion models and datasets.** In line with prior works on memorization in diffusion models [40, 49, 34, 19, 8], we evaluate our proposed methods using Stable Diffusion v1.4 [35]. Although Stable Diffusion v2.0 exhibits fewer memorization issues due to being trained on a de-duplicated dataset, it still suffers from template memorization—a form of memorization in which a set of prompts leads to images that are highly similar to training samples, differing mainly in aspects such as color or style [34]. To account for this, we also employ Stable Diffusion v2.0 in our evaluations. For Stable Diffusion v1.4, we use 500 memorized prompts extracted from the LAION dataset [38], and for Stable Diffusion v2.0, we use 219 memorized prompts. These prompts are provided by Webster [47] and Ren et al. [34], respectively.

**Evaluation metrics.** We employ four complementary metrics to evaluate memorization, image-text alignment, image quality, and image diversity. To assess the degree of memorization in generated images, we use the Self-Supervised Copy Detection (SSCD) score [32], which measures object-level similarity between a generated image and its nearest neighbor in the training set. Image-text alignment is assessed with the CLIP score [33], which quantifies the semantic consistency between each generated image and its corresponding text prompt. For image quality and prompt alignment, we adopt ImageReward [50], while image diversity is evaluated using LPIPS [52]. Lower SSCD scores indicate stronger mitigation of memorization, whereas higher CLIP, LPIPS, and ImageReward scores reflect better performance. Detailed evaluations of image quality and diversity are presented in Appendix D.

**Baselines.** We compare our proposed methods against six baselines. The first is a no-mitigation baseline, where images are generated using Stable Diffusion without applying any memorization mitigation. Additionally, we consider five inference-time mitigation approaches: adding random tokens [40], adjusting the prompt embedding [49], scaling cross-attention logits [34], applying opposite guidance during the early stages of sampling [22], and applying guidance only after a specified timestep [22]. All baseline methods are applied at inference time and do not require access to the training data, making them suitable for fair comparison with our proposed approaches.

**Implementation details.** For each diffusion model, we generate 10 images per memorized prompt using identical inference configurations across all baselines and our proposed methods. Specifically, for Stable Diffusion v1.4, we employ a DDIM sampler [41] with 50 sampling steps and a CFG scale of 7, whereas for Stable Diffusion v2.0, we use a Euler discrete scheduler [26] with the same number of steps and CFG scale. Further implementation details are provided in Appendix B.

## 4.2 Impact of initial sample adjustment on memorization mitigation

To examine the effectiveness of our proposed initial sample adjustment in mitigating memorization while promoting image-text alignment, we compare mitigation results obtained with and without adjustment. The adjustment is implemented using the Batch-wise method. For the unadjusted case, the starting timestep for applying CFG, denoted as $\tau$, is varied as $\tau \in \{780, 760, 740, 720\}$, while for the adjusted case it is set to $\tau \in \{900, 880, 860, 840\}$. This design ensures that we can isolate the impact of the adjustment itself, given that $\tau$ influences mitigation performance. The results are illustrated in Figure 4. On the No adjustment curve, the rightmost and leftmost data points correspond to $\tau = 780$ and

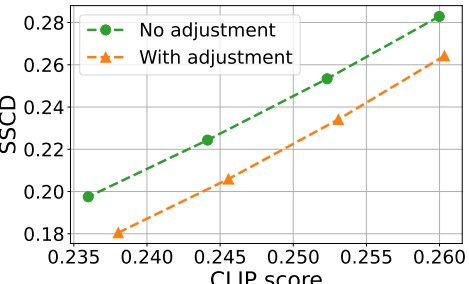

Figure 4: Comparison of SSCD and CLIP scores with and without initial sample adjustment.

$\tau = 720$, respectively; for the With adjustment curve, they correspond to $\tau = 900$ and $\tau = 840$. Despite the earlier application of CFG (*i.e.*, higher $\tau$), the adjusted cases consistently achieve lower SSCD scores than their unadjusted counterparts at comparable CLIP scores, indicating that the proposed adjustment effectively shifts the transition point earlier in the denoising process and enables memorization mitigation even when CFG is applied early. More importantly, the adjustment yields a more favorable trade-off between SSCD score and CLIP score. These results demonstrate that our initial sample adjustment effectively reduces memorization while encouraging image-text alignment.

## 4.3 Comparison with baselines

In this subsection, we compare our proposed methods with baseline approaches by analyzing how SSCD and CLIP scores vary across different mitigation strengths, following prior works [34, 49]. We evaluate both the baselines and our Per-sample method under five distinct hyperparameter settings. It is worth noting that mitigation strength in our methods is closely tied to the value of $||\tilde{\epsilon}_\theta(x_T, T, y)||_2$. To control this value, the Batch-wise method employs three hyperparameters ($\tilde{\gamma}, \rho$, and $M$), whereas the Per-sample method uses only one ($l_{\text{target}}$). Since both approaches share the core idea that adjusting the initial noise $x_T$ to minimize $||\tilde{\epsilon}_\theta(x_T, T, y)||_2$ improves mitigation performance, we report the performance of Per-sample across multiple $l_{\text{target}}$ values. Detailed hyperparameter configurations

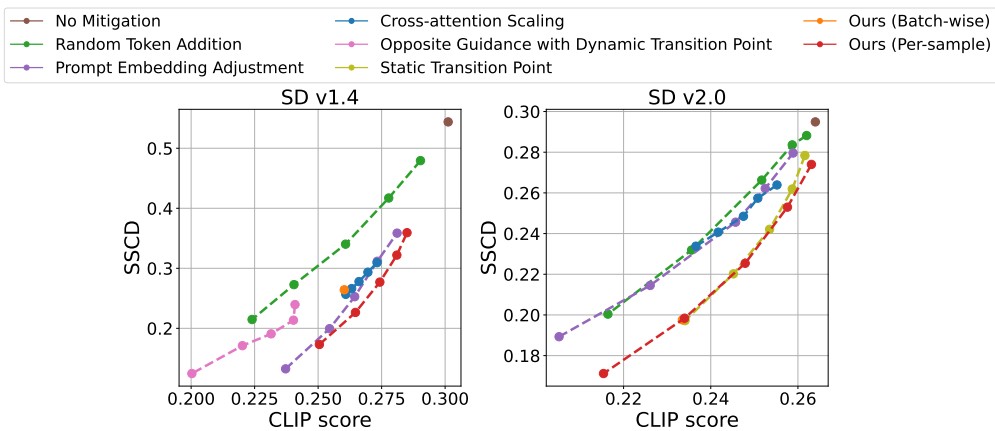

Figure 5: Comparison of SSCD and CLIP scores among different mitigation methods under Stable Diffusion v1.4 and v2.0. Lower SSCD scores indicate stronger memorization mitigation, while higher CLIP scores indicate better image-text alignment.

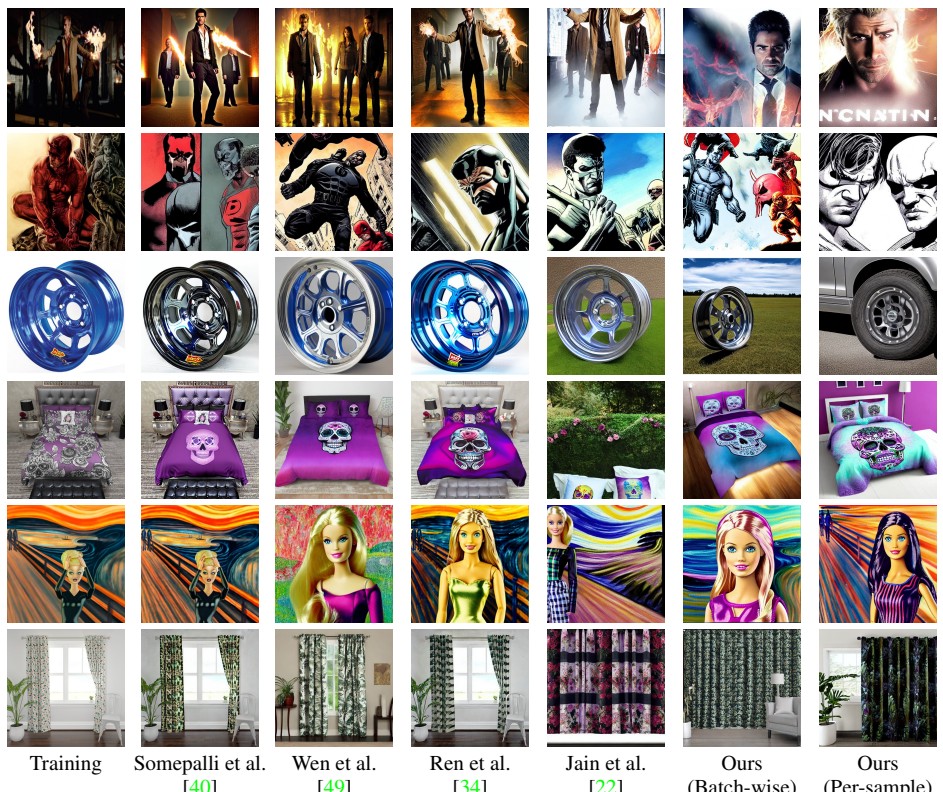

Figure 6: Qualitative comparison of memorization mitigation results. Each column shows generations produced by various baseline methods and our proposed approaches. The prompts used for the generations are provided in Appendix G.

for all methods are provided in Appendix B. The results in Figure 5 show that Batch-wise achieves performance comparable to Cross-attention Scaling under Stable Diffusion v1.4, while achieving the best performance under v2.0. More importantly, Per-sample consistently provides the most favorable SSCD–CLIP trade-off across both diffusion models, highlighting its superior effectiveness in mitigating memorization. Qualitative comparisons are presented in Figure 6.

A performance gap is observed between the Batch-wise and Per-sample methods under Stable Diffusion v1.4. We conjecture that this difference arises because the optimal hyperparameters for

the Batch-wise method vary across prompts and initial samples. As shown in Figure 3, a fixed set of hyperparameters can lead to different degrees of magnitude reduction in $\tilde{\epsilon}_\theta(x_T, T, y)$, resulting in varying transition points depending on the prompt and the initial sample. Moreover, since CFG must be applied after—but not far beyond—the transition point, the optimal timestep for CFG application is inherently sample-specific in the Batch-wise setting. In contrast, the Per-sample method directly minimizes the magnitude of $\tilde{\epsilon}_\theta(x_T, T, y)$ via backpropagation with fixed hyperparameters, effectively eliminating the transition point for each individual generation. This enables robust memorization mitigation while improving image-text alignment, as demonstrated in Figure 5. Although the Batch-wise approach is less adaptive and slightly underperforms relative to the Per-sample method, it still achieves competitive performance.

# 5 Related work

**Memorization in diffusion models.** Beyond supervised models [4] and language models [3], diffusion models [5, 39, 40] have also been found to exhibit a memorization problem, indicating that models tend to memorize and regenerate training data. This phenomenon raises significant privacy concerns, as it can result in unintended data leakage. To address this issue, various mitigation strategies have been explored. While early approaches sought to mitigate memorization either during training or at inference, training-time mitigation typically requires access to the original dataset and substantial computational resources, limiting its practicality. Consequently, recent efforts have shifted toward inference-time mitigation strategies [7, 22, 19, 8, 34]. For example, Wen et al. [49] proposes adjusting prompt embeddings to minimize the magnitude of the conditional noise prediction, while Chen et al. [8] introduces attention masking to enhance this adjustment. Somepalli et al. [40] mitigates memorization by perturbing prompts through token addition or replacement, and Jain et al. [22] suggests applying opposite guidance during early denoising steps to counteract memorization induced by CFG. Our work follows this line of research, proposing inference-time mitigation strategies that adjust initial noise samples to reduce memorization.

**Impact of initial noise samples.** Recent studies have increasingly focused on the impact of initial noise samples on image generation in diffusion models, exploring how modifying these samples can address various challenges, such as improving image-text alignment and enhancing image quality [51, 31, 17, 44, 46, 6, 37, 9]. For example, Wallace et al. [45] proposes an end-to-end framework that backpropagates through the entire diffusion process to adjust the initial noise sample, thereby improving classifier guidance. Guo et al. [18] utilizes attention maps to guide the refinement of initial noise samples, aiming to mitigate semantic errors such as subject neglect, subject mixing, and incorrect binding. Similarly, Eyring et al. [15] addresses poor semantic fidelity in text-to-image models by optimizing the initial noise samples based on reward feedback.

**Sharpness aware minimization.** The sharpness term used to minimize the magnitude of the conditional noise prediction in our method was proposed in Sharpness-Aware Minimization (SAM) [16]. SAM was originally designed to seek flatter minima in the loss landscape, motivated by several studies showing that flatter minima are associated with better model generalization [27, 25, 14]. By simultaneously minimizing both the loss value and the sharpness of the loss surface, SAM aims to find model parameters that reside in regions where the loss values remain consistently low. SAM has achieved state-of-the-art generalization performance, resulting in numerous follow-up works that aim to improve it by introducing surrogate loss function [54], proposing geometric measures of neighborhood [29, 28], or enhancing training efficiency [12, 13, 24]. However, a deep understanding of which components of SAM drive its generalization improvement remains limited. As a result, there have been significant efforts to analyze the underlying mechanisms of SAM [1, 48, 10, 53]. For instance, Zhao et al. [53] reveals that SAM effectively minimizes the gradient norm during optimization.

In contrast to previous works, to the best of our knowledge, this paper is the first to investigate the impact of the initial noise sample on mitigating memorization in diffusion models. By introducing a novel method that adjusts the initial sample by minimizing the magnitude of the conditional noise prediction, we analyze the relationship between the magnitude of conditional guidance and the transition point. Building on this analysis, we propose two novel mitigation strategies that effectively reduce memorization while promoting image-text alignment.

## 6 Limitations

While our proposed methods demonstrate strong effectiveness, they also have certain limitations. The Per-sample method introduces a slight increase in computational overhead, and the performance of the Batch-wise method can vary with hyperparameter selection. However, we believe that the additional cost of the Per-sample method is well justified by its effectiveness in mitigating memorization and enhancing privacy and intellectual property protection. Moreover, the close relationship between the hyperparameters of the Batch-wise method and the goal of reducing the magnitude of conditional guidance offers a clear and efficient direction for hyperparameter tuning.

## 7 Conclusion

We present a novel perspective on the role of initial samples in mitigating memorization in diffusion models, viewed through the lens of the attraction basin. We observe that different initializations result in different transition points. Given that earlier transition points lead to non-memorized images that remain well-aligned with text prompts, we further analyze which initial samples facilitate faster escape from the attraction basin and find that adjusting initial noise samples to achieve a smaller magnitude of the conditional noise prediction results in earlier transition points. Building on these insights, we propose two novel inference-time mitigation strategies—Batch-wise and Per-sample approaches—which adjust initial samples collectively or individually and use them for image generation. Experimental results validate our findings and demonstrate the effectiveness of the proposed strategies in mitigating memorization while improving image-text alignment.

## Acknowledgements

This work is in part supported by the National Research Foundation of Korea (NRF, RS-2024-00451435(20%), RS-2024-00413957(20%)), Institute of Information & communications Technology Planning & Evaluation (IITP, RS-2021-II212068(10%), RS-2025-02305453(10%), RS-2025-02273157(10%), RS-2025-25442149(10%), 2021-0-00180(10%), RS-2021-II211343(10%)) grant funded by the Ministry of Science and ICT (MSIT), Institute of New Media and Communications(INMAC), and the BK21 FOUR program of the Education, Artificial Intelligence Graduate School Program (Seoul National University), and Research Program for Future ICT Pioneers, Seoul National University in 2025.

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

# Appendix

## A    Pseudocode for Per-sample mitigation

We present the pseudocode for the Per-sample mitigation method in Figure 7.

```python
def adj_latent(latent, t, prompt_embeds, lr, target_loss):
    latent.requires_grad = True
    optim = AdamW([latent], lr=lr)
    while True:
        pred_uncond, pred_text = unet(latent, t, prompt_embeds)
        loss = norm(pred_text - pred_uncond)
        if loss < target_loss:
            break
        optim.zero_grad(); loss.backward(); optim.step()
    return latent.detach()

def per_sample_mitigation(prompt_embeds, cfg_scale, lr, target_loss):
    latent = randn(latent_shape)
    for i, t in enumerate(timesteps):
        # Adjust the initial sample using backpropagation
        if i == 0:
            latent = adj_latent(latent, t, prompt_embeds, lr, target_loss)
        # Perform the denoising process with the adjusted initial sample
        pred_uncond, pred_text = unet(latent, t, prompt_embeds)
        noise_pred = pred_uncond + cfg_scale * (pred_text - pred_uncond)
        latent = scheduler.step(noise_pred, t, latent)
    return latent
```

Figure 7: Pseudocode illustrating the Per-sample mitigation method.

## B    Implementation details

All experiments were conducted using an NVIDIA A100 GPU. The implementation environment included Hugging Face 0.23.4, CUDA 12.0, Python 3.8.0, and PyTorch 2.3.1 with Torchvision 0.18.1. Detailed configurations for the proposed mitigation strategies and baselines are outlined below.

**Batch-wise mitigation.**    For Stable Diffusion v1.4, the adjustment strength is set to $\tilde{\gamma} = 0.7$, the sharpness parameter to $\rho = 50.0$, and the number of adjustments to $M = 2$. CFG is applied with a guidance scale of 7.0 for timesteps $t \leq \tau$, and disabled (*i.e.*, set to 0.0) for $t > \tau$, where the CFG application start timestep is set to $\tau = 900$. For Stable Diffusion v2.0, we set $\tilde{\gamma} = 0.7$, $\rho = 30.0$, $M = 4$, and $\tau = 917.4490$.

**Per-sample mitigation.**    For Stable Diffusion v1.4, the initial noise sample is adjusted using the AdamW optimizer [30] with a learning rate of 0.01, while all other hyperparameters remain at their default values. The target threshold is varied as $l_{\text{target}} \in \{0.7, 0.9, 1.1, 1.3, 1.5\}$. For Stable Diffusion v2.0, the initial noise sample is adjusted using the same optimizer with a learning rate of 0.1, and the target threshold is varied as $l_{\text{target}} \in \{5, 8, 10, 15, 20\}$.

**Baselines.**    For Random Token Addition, the number of added tokens is set to $\{1, 2, 4, 6, 8\}$ for both diffusion models. For Prompt Embedding Adjustment, the target loss is set to $l_{\text{target}} \in \{3, 4, 5, 6, 7\}$ for Stable Diffusion v1.4 and $l_{\text{target}} \in \{50, 60, 70, 80, 90\}$ for Stable Diffusion v2.0. For Cross-attention Scaling, the re-scaling factor is set to $C \in \{1.15, 1.2, 1.25, 1.3, 1.35\}$ for both diffusion models. For Opposite Guidance with Dynamic Transition Point, the opposite guidance scale is

set to $\{1, 3, 5, 7, 9\}$. For Static Transition Point, the CFG application start timestep is set to $\tau \in \{999.0, 978.6122, 958.2245, 937.8367, 917.4490\}$.

## C  Analysis of distributional shifts induced by initial noise adjustment

We observed that using initial noise with weaker conditional guidance leads to an earlier transition point, thereby improving memorization mitigation performance. Consequently, our objective is to effectively reduce the norm $||\tilde{\epsilon}_\theta(x_T, T, y)||_2$. A natural question arises: is it appropriate to minimize this norm without an explicit regularization term to prevent the adjusted initial noise from becoming out-of-distribution (OOD)? We argue that the weight decay in the AdamW optimizer partially fulfills this role. Furthermore, with the default weight decay value of $w = 0.01$, we observed no degradation in either mitigation performance or image quality. The following subsections provide both theoretical and empirical evidence supporting this claim.

### C.1  Effect of weight decay in mitigating distributional shift

According to prior works [2, 36], the L2 norm of the initial noise $x_T \sim \mathcal{N}(\mathbf{0}, \mathbf{I})$ follows a chi distribution:

$$||x_T||_2 = \sqrt{\sum_{i=1}^{d} {x_T^i}^2} \sim \chi^d = ||x_T||_2^{d-1} e^{-||x_T||_2^2/2} / (2^{d/2-1} \Gamma(\frac{d}{2})), \tag{11}$$

where $d$ denotes the dimensionality of $x_T$ and $\Gamma(\cdot)$ is the Gamma function. Based on this, to prevent the adjusted initial noise from deviating into an OOD state, one could consider minimizing the objective

$$||\tilde{\epsilon}_\theta(x_T, T, y)||_2 - w \log p(||x_T||_2), \tag{12}$$

where $p$ represents the chi distribution and $w$ is a weighting coefficient. By substituting the chi distribution from Equation (11) into $p(||x_T||_2)$ in Equation (12), the objective can be rewritten as

$$||\tilde{\epsilon}_\theta(x_T, T, y)||_2 + c - w(d-1) \log ||x_T||_2 + w \frac{||x_T||_2^2}{2}, \tag{13}$$

where $c$ is a constant independent of $x_T$. We argue that the weight decay term in the AdamW optimizer implicitly minimizes the final quadratic term, $\frac{||x_T||_2^2}{2}$, thereby constraining adjusted initial samples from becoming OOD. To empirically validate this, we compared 5000 unadjusted and 5000 adjusted initial noise samples, measuring the Jensen–Shannon divergence (JSD) between their distributions under varying weight decay values. The results shown in Table 1 demonstrate that as $w$ increases, JSD decreases. This suggests that stronger weight decay encourages the adjusted samples to remain closer to the original noise distribution.

Table 1: Jensen–Shannon divergence between adjusted and unadjusted initial noise distributions under varying weight decay values.

|  | $w = 0.01$ | $w = 0.05$ | $w = 0.1$ | $w = 0.2$ |
|---|---|---|---|---|
| JSD | 0.0665 | 0.0663 | 0.0638 | 0.046 |

### C.2  Effect of weight decay on mitigation performance

We varied the weight decay parameter $w$ as described in the previous subsection and evaluated the corresponding performance. The results are presented in Table 2. In this experiment, the target threshold was fixed at $l_{\text{target}} = 0.9$. Across all tested settings, the performance remained largely consistent regardless of the value of $w$, suggesting that the default weight decay value $w = 0.01$ does not cause degradation in either mitigation performance or image quality. We attribute this stability to the fact that the default value does not cause the adjusted initial noise to deviate significantly from the original distribution $\mathcal{N}(\mathbf{0}, \mathbf{I})$. This interpretation is further supported by the JSD results in Table 1, where a JSD value of 0.0665 at $w = 0.01$ indicates minimal distributional shift.

Table 2: Performance comparison across different weight decay values.

|  | $w = 0.01$ | $w = 0.05$ | $w = 0.1$ | $w = 0.2$ |
|---|---|---|---|---|
| SSCD | 0.2265 | 0.2265 | 0.2265 | 0.2275 |
| CLIP | 0.2647 | 0.2647 | 0.2647 | 0.2657 |
| LPIPS | 0.7615 | 0.7615 | 0.7615 | 0.7610 |
| ImageReward | -0.3522 | -0.3522 | -0.3522 | -0.3501 |

## D   Performance comparison on image quality and diversity

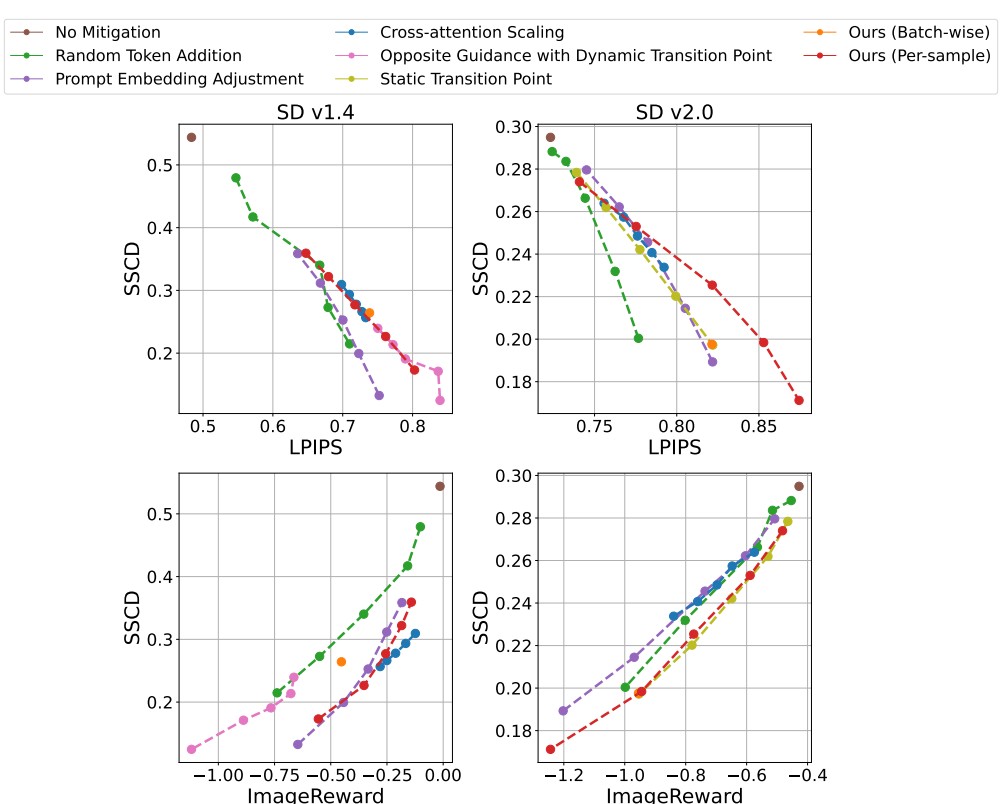

Figure 8: Comparison of different mitigation methods under Stable Diffusion v1.4 and v2.0. The first row shows SSCD versus LPIPS, while the second row shows SSCD versus ImageReward. Lower SSCD scores indicate stronger memorization mitigation, whereas higher LPIPS and ImageReward values reflect greater image diversity and quality.

We compare our proposed methods with baseline approaches by analyzing how LPIPS and ImageReward values vary across different mitigation strengths. To this end, we present two sets of results: SSCD versus LPIPS and SSCD versus ImageReward, using the same hyperparameters as in Figure 5. The results are shown in Figure 8. Since effective memorization mitigation enables the generation of non-memorized images, it also promotes greater diversity in the outputs, as illustrated in the SSCD–LPIPS results in the first row. Under Stable Diffusion v1.4, the Batch-wise method achieves the highest LPIPS values at comparable SSCD scores, while Per-sample achieves the second-highest LPIPS values at lower SSCD scores. In contrast, under Stable Diffusion v2.0, Batch-wise yields the second-highest LPIPS values, whereas Per-sample attains the highest LPIPS values with a substantial margin. The SSCD–ImageReward results in the second row further show that the Per-sample approach provides the most favorable trade-off between memorization mitigation and image quality across both diffusion models. These findings demonstrate that our proposed methods introduce minimal quality degradation while substantially enhancing image diversity, thereby improving the practical utility of diffusion models.

# E    Magnitude of conditional noise prediction across varying adjustment strengths

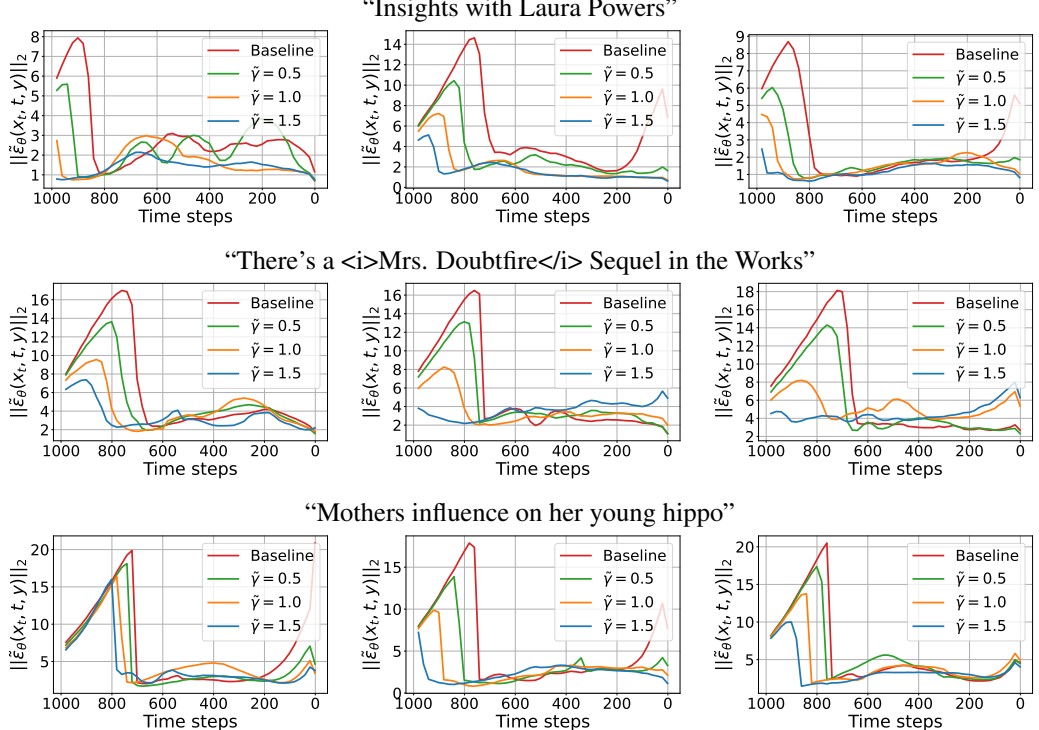

Figure 9: Magnitude of $\tilde{\epsilon}_\theta(x_t, t, y)$ at each timestep during sampling without CFG under varying adjustment strengths $\tilde{\gamma}$. Each row corresponds to a different memorized prompt, and each column corresponds to a distinct initial noise sample.

We provide additional plots in Figure 9 illustrating the magnitude of conditional guidance at each timestep during denoising without CFG, under varying adjustment strengths $\tilde{\gamma}$. The results show a consistent trend: as $\tilde{\gamma}$ increases, the magnitude of $\tilde{\epsilon}_\theta(\tilde{x}_T, T, y)$ decreases, leading to an earlier occurrence of the transition point across different prompts and initial noise samples.

# F    Magnitude analysis of conditional guidance under Per-sample mitigation

In the Per-sample method, we adjust the initial noise sample $x_T$ by minimizing the magnitude of $\tilde{\epsilon}_\theta(x_T, T, y)$ until the transition point effectively disappears. To validate this effect, we compare the magnitude of the conditional noise prediction with and without applying the Per-sample method. In these experiments, we set the target loss $l_{\text{target}} = 0.9$ and use a learning rate of 0.01. Figure 10 presents the results: each row corresponds to a different memorized prompt, and each column corresponds to a different initial noise sample $x_T$. The Per-sample method significantly reduces the magnitude of $\tilde{\epsilon}_\theta(x_T, T, y)$. Moreover, unlike the baseline, no sharp drop in magnitude is observed, indicating that the transition point has been effectively eliminated. This enables CFG to be applied from the beginning of the denoising process.

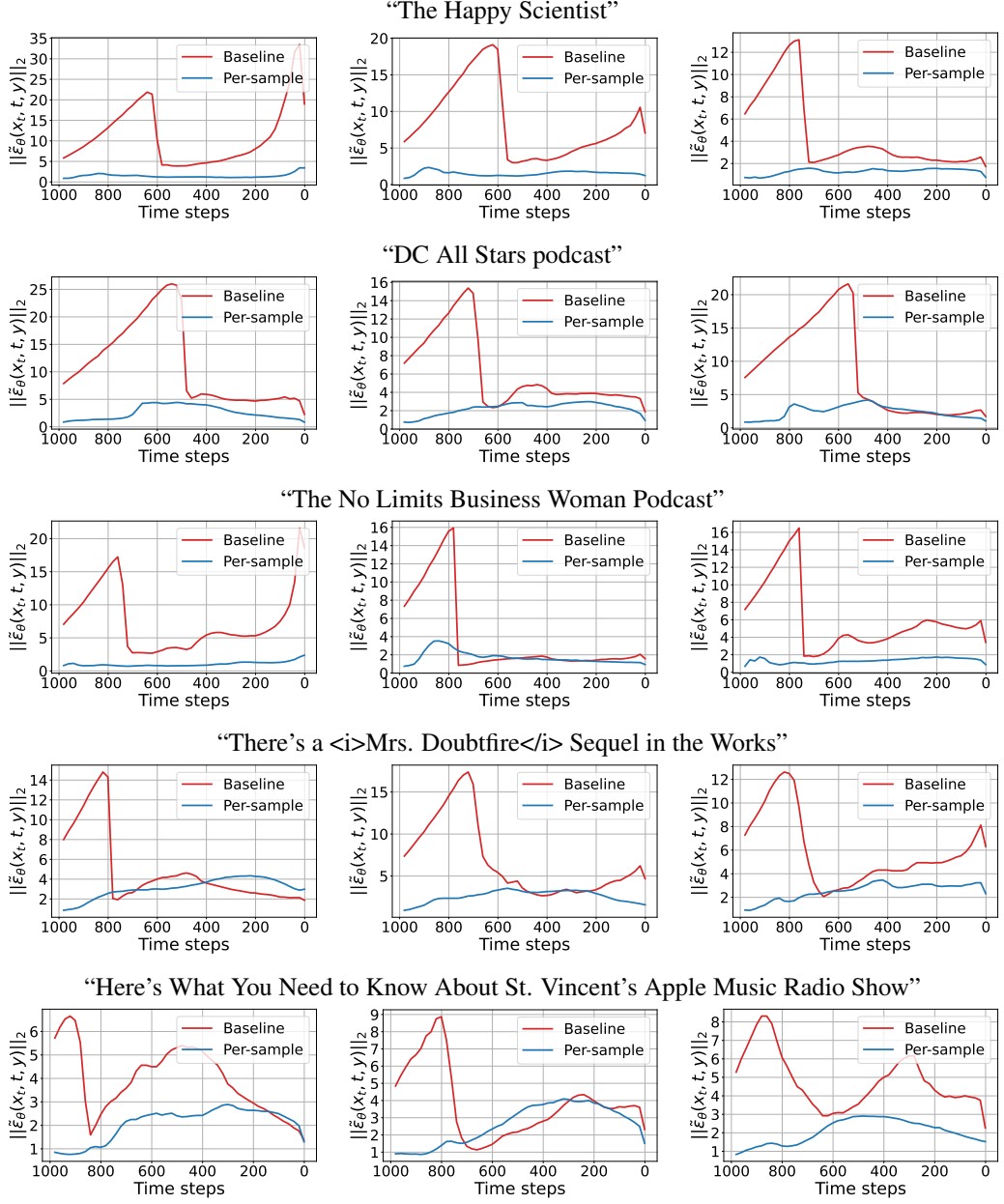

Figure 10: The magnitude of the conditional noise prediction at each timestep during sampling without CFG for five memorized prompts. The baseline corresponds to the case without adjustment, while the Per-sample method is applied with a target loss $l_{\text{target}} = 0.9$.

## G  Prompts for Figure 6

The following memorized prompts were used to generate the samples shown in Figure 6:

- Watch the Trailer for NBC's Constantine
- As Punisher Joins Daredevil Season Two, Who Will the New Villain Be?
- Aero 50-975035BLU 50 Series 15x7 Inch Wheel, 5 on 5 Inch BP 3-1/2 BS
- Signature Purple Ombre Sugar Skull and Rose Bedding
- If Barbie Were The Face of The World's Most Famous Paintings
- Plymouth Curtain Panel featuring Madelyn - White Botanical Floral Large Scale by heatherdutton

# H Additional qualitative results

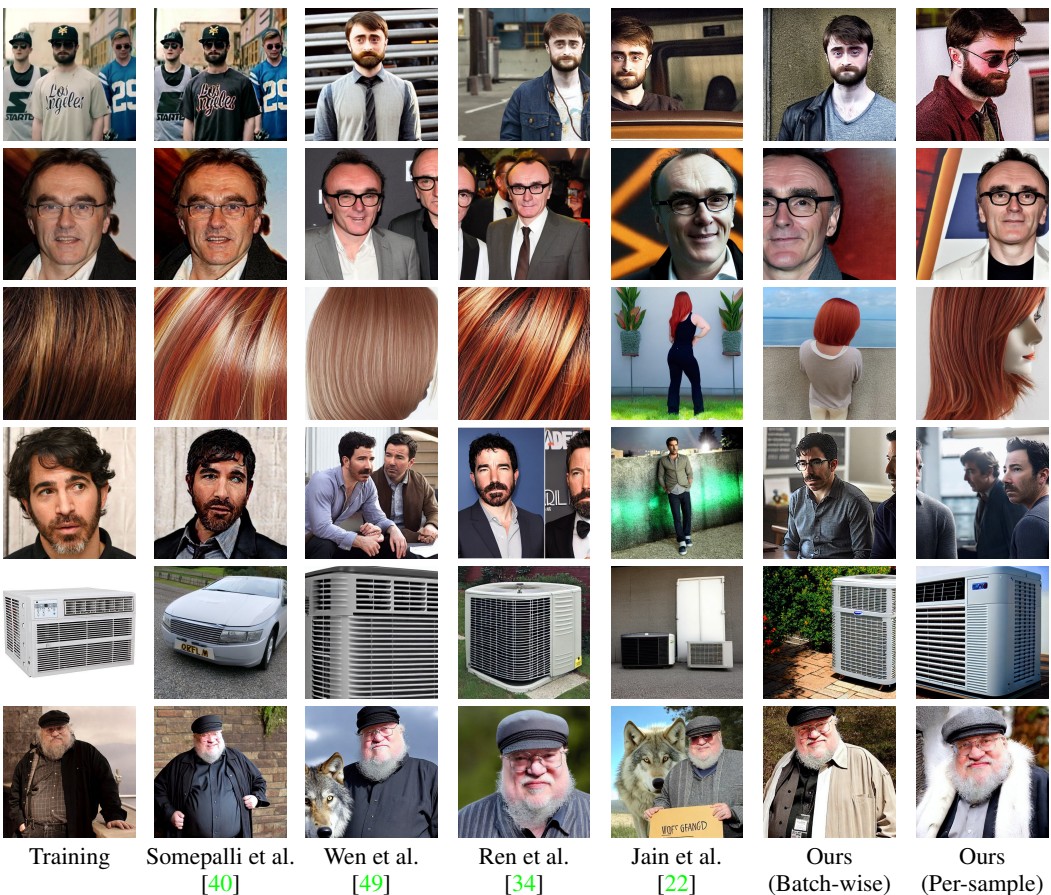

Figure 11: Additional qualitative comparison between our proposed methods and baseline approaches. Each column presents images produced by a baseline or one of our methods, with the leftmost column showing the corresponding training images for reference.

We present additional qualitative results comparing the baseline methods with our proposed approaches in Figure 11. The memorized prompts used to generate these samples are listed below:

- Daniel Radcliffe Dons a Beard and Saggy Jeans in Trailer for BBC GTA Miniseries The Gamechangers
- Director Danny Boyle Is Headed To TV With FX Deal
- 30A27S4 - Shaded Peach - Brown Red/Strawberry Blonde Blend, Shaded w/ Dk Brown
- Chris Messina In Talks to Star Alongside Ben Affleck in Live By Night
- Air Conditioners & Parts
- George R.R. Martin Donates $10,000 to Wolf Sanctuary for a 13-Year-Old Fan

# I Computational cost analysis

Table 3: Computational cost of different mitigation methods.

|  | gpu memory usage (MB) | inference time (sec) |
|---|---|---|
| No Mitigation | 3564.94 | 1.85 |
| Random Token Addition | 3573.57 | 1.85 |
| Prompt Embedding Adjustment | 8115.53 | 1.98 |
| Cross-attention Scaling | 3170.38 | 1.93 |
| Opposite Guidance with Dynamic Transition Point | 3573.57 | 1.87 |
| Ours (Batch-wise) | 3573.57 | 2.01 |
| Ours (Per-sample) | 9701.62 | 2.8 |

We conducted an additional experiment to compare baseline mitigation methods with our proposed approaches in terms of computational cost. For a fair comparison, we measured GPU memory consumption and inference time per image. The results, presented in Table 3, show that the Batch-wise method introduces negligible computational overhead, while the Per-sample method requires relatively higher memory usage and longer inference time. However, as demonstrated in Figure 5, the Per-sample method consistently achieves superior memorization mitigation and robustness across different models. Given that effective mitigation is crucial for enhancing privacy and safeguarding intellectual property while reducing potential risks, we believe that the additional computational cost of the Per-sample approach is a reasonable trade-off.

