# OpenReview forum: "Adjusting Initial Noise to Mitigate Memorization in Text-to-Image Diffusion Models"
_NeurIPS.cc/2025/Conference — NeurIPS 2025 poster_

### Official Review · Reviewer_tNu7 · 2025-06-14

**Clarity:** 3
**Significance:** 2
**Originality:** 3
**Rating:** 3
**Confidence:** 3

**Summary:**

The paper investigates how to stop diffusion models from recreating train-data images. It builds upon the framework of attraction cones and proposes that the initial noise can be shifted to avoid falling into attraction cones. This allows to stay outside the memoization cones from an earlier point on, so that CFG can be applied almost from start (in the case of the single-sample method even from start, though at higher computational cost), which retains prompt-adherence better.

**Questions:**

Section 3.1.2. currently relies only on a qualitative analysis. Could you add a quantitative study across a whole dataset?

Currently, your analyses focus on 1000-step diffusions. How does it look like for 5-20 step diffusions (for example on SD3 (which you already use) or FLUX (just if you have time to test it; it uses the same schedulers etc as SD3, so it should be plug-and-play if you load it via huggingface))?

Currently, you base the stop-point analysis on the timestep index. However, this implicitly depends on which scheduler you use. Could you have a look at how it looks with the Signal-to-noise ratio on the x-axis? (as suggested by “Understanding Diffusion Objectives as the ELBO with Simple Data Augmentation“, 2023)

Given that you find that the initial noise matters — can you identify any regions in the initial noise space? If we could sample from that region from the get-go, we could prevent falling into attraction cones.

Do you have an intuition of how far your initial noise adjustment pushes the noise away? This could give a rough estimate of how large the diameter of the attraction cones is at t=T, so how much of the initial noise Gaussian is affected by memorization.

Is there any runtime difference between the batchwise mitigation and the per-sample mitigation? It seems like the only difference is whether we use pooled backprop gradients or per-sample ones (please correct me if I’m wrong), so probably single-sample only uses more VRAM to store the individual gradients?

Could you try out combining your method with orthogonal methods like prompt embedding adjustment? It would be interesting to see if they are truly orthogonal (and thus add performance in combination) or doing the same by different means (no performance boost by combining them). It would be best to plot pareto fronts.

How was the CFG weight of 7 chosen? In my experience, modern diffusion models usually have lower CFGs to, e.g., give the best FID.

**Ethical Concerns:**

["NO or VERY MINOR ethics concerns only"]

**Final Justification:**

After the rebuttal discussion and having read the rebuttal discussions with all other reviewers, I think the score of 3 still applies to the paper. The method gives an interesting view into the diffusion dynamics, but the results are thin: All evaluations are on (10x) 500 prompts as opposed to the typical 50k, and models include only SD 1.4, and now with the rebuttal SD 2.0, although more modern diffusion models already exist (even with the same Euler Discrete Scheduler, so no code changes are needed), and usually behave much smoother in terms of memorization prevention. What adds to my scepticism is that in the rebuttal, when providing the results on SD 2.0, the authors claim "Per-sample achieves the best Pareto front" but when plotting the results it is obvious that the Pareto front is exactly overlaying with a very simple baseline. These points leave me sceptical with the paper's sometimes bold claims.

I believe the responsibility of a reviewer towards the field is to be willing to defend the paper should future studies find it to not reproduce, and with the current thin evaluation I am not convinced enough to to so. As such, I believe the most secure way forward is to reject at this point, but keep working on the paper a few more weeks to make the findings more robust by analyzing modern Diffusion models, not shying back from admitting when the method does _not_ work to give an honest scientific insight into diffusion dynamics, and then resubmit to, e.g., ICLR.

**Limitations:**

The authors discuss the runtime overhead and hyperparamter tuning in an explicit Limitations section. The fact that SD2 (and above) have close to no problems with exact memorization is discussed, but outside the limitations section.

**Paper Formatting Concerns:**

None.

**Quality:**

2

**Strengths And Weaknesses:**

## Strengths

* The problem at hand, memorization is well-motivated and relevant
* The authors nicely first introduce the principles and some proof-of-works of the method before applying it on large scale
* The used metrics are reasonable, and the authors measure both the desired memorization but also impacts on prompt adherence (CLIP Score) and image quality (FID)

## Weaknesses

### Weaknesses that influence my score, in order of strength:

* The motivation for the work — that diffusion models memorize exact images — might be a problem of the past. The authors note that they use Stable Diffusion v1.4 because SD2 already does not have the problem anymore (let alone SD3). This is in line with my experimental experience. Also, the CFG strength is quite strong (=7), probably at more modest strengths there is less memorization to begin with. This limits the relevance of the work, but I still find it interesting to investigate the effect, e.g., for diffusion models that were trained on less data (even in the future).
* The evaluation of the work is very thin. The method is tested on one model (SD v1.4) on 500 prompts (generating 10 images each). Since the authors already note that more modern Diffusion models have less memorization problems, it would be interesting to see results across different scales (SD2, SD3) and architectures (FLUX, RGB-space diffusion) to understand it better and to show the robustness of the proposed method.
* According to Table 1, the proposed method is not much better than prompt embedding adjustment. This is a slight performance-weakness, but in my view not a complete rejection reason, because prompt adjustment and noise adjustment are orthogonal approaches, and in some approaches prompt embedding adjustment might not be available (unconditional generation) or desired (might be attached with adversarial prompting).
* Table 1 currently shows results for only one hyperparameter, and it is not explained how that was chosen. The results would be more trustworthy if there were Pareto fronts across different hyperparameters.
* There are three hyperparamters: two to control how strongly to apply mitigation, one to control how many mitigation steps to make (plus one to decide when to start CFG, but other methods also have this one, and it is removed in the single-sample method)

### Other notes (that do not influence my score but could improve the camera-ready)

* It might be interesting to add FD_Dinov2 as an evaluation metric, if computationally feasible, because it is more detailed than FID (based on Inception v3).

## Justification for score

The paper is logically structured and the method reasonable in theory and performant in practice. However, there are two bigger issues: First, the experimental evaluation is very thin with only one model and 500 prompts. Second, the tested model is quite old (SD1.4) because more modern diffusion models have less memorization problems to begin with. This gives this paper more of an impression of an initial proof-of-concept suitable for a workshop rather than a full conference paper. I am willing to increase my score if the authors can improve the experimental evaluation in particular.

---

> ### Author Response · Authors · 2025-07-31
>
> Thank you for your insightful comments. We've carefully considered them and provided responses. Please let us know if you need any clarification or have additional questions.
>
> > **W1**: On the motivation for the work and the CFG strength.
>
> - The motivation for the work
>
> As noted in Section 4.1, Stable Diffusion v2 shows fewer memorization issues, but they are not entirely resolved. Prior works [1, 2] confirm that while verbatim memorization has decreased, template memorization remains common.
>
> Given the potential risks of leaking sensitive or copyrighted content (Section 1), even rare cases of memorization are concerning. Addressing them remains an important objective.
>
> - On the CFG strength
>
> As you noted, lowering CFG strength can reduce memorization, but the reduction is modest. We evaluated this using CFG strengths {2, 3, 5, 7}, with results shown in the table below.
>
> While SSCD scores decrease at lower strengths, the effect is limited—for example, CFG = 2 still shows substantial memorization (SSCD = 0.4908).
>
> Importantly, applying Per-sample consistently leads to much lower SSCD scores across all CFG values, demonstrating its robustness and effectiveness.
>
> |                                | SSCD   | CLIP   |
> |--------------------------------|--------|--------|
> | No Mitigation (cfg strength=7) | 0.5439 | 0.3012 |
> | No Mitigation (cfg strength=5) | 0.5409 | 0.3005 |
> | No Mitigation (cfg strength=3) | 0.526  | 0.294  |
> | No Mitigation (cfg strength=2) | 0.4908 | 0.2813 |
> | Per-sample (cfg strength=7)    | 0.2265 | 0.2647 |
> | Per-sample (cfg strength=5)    | 0.1945 | 0.2556 |
> | Per-sample (cfg strength=3)    | 0.1456 | 0.2335 |
> | Per-sample (cfg strength=2)    | 0.1107 | 0.2087 |
>
> > **W2 & Q2**: Performance results under SDv2.0.
>
> To demonstrate the robustness and generalizability of our proposed method, we conducted experiments using SDv2.0 and the Euler discrete scheduler. The performance results are shown in the table below, where Per-sample achieves the best Pareto front. Batch-wise also performs comparably well.
>
> |                                           | SSCD   | CLIP   |
> |-------------------------------------------|--------|--------|
> | No Mitigation                             | 0.2949 | 0.264  |
> | Adding 2 Random Tokens                    | 0.2836 | 0.2587 |
> | Adding 4 Random Tokens                    | 0.2663 | 0.2517 |
> | Adding 6 Random Tokens                    | 0.2319 | 0.2356 |
> | Adding 8 Random Tokens                    | 0.2004 | 0.2164 |
> | Prompt Embedding Adjustment (l_target=50) | 0.1893 | 0.2052 |
> | Prompt Embedding Adjustment (l_target=60) | 0.2145 | 0.2261 |
> | Prompt Embedding Adjustment (l_target=70) | 0.2456 | 0.2457 |
> | Prompt Embedding Adjustment (l_target=80) | 0.2622 | 0.2525 |
> | Cross-attention Scaling (c=1.2)           | 0.2574 | 0.2508 |
> | Cross-attention Scaling (c=1.25)          | 0.2485 | 0.2475 |
> | Cross-attention Scaling (c=1.3)           | 0.2407 | 0.2417 |
> | Cross-attention Scaling (c=1.35)          | 0.2338 | 0.2366 |
> | Static Transition Point (tau=978.6122)    | 0.2619 | 0.2587 |
> | Static Transition Point (tau=958.2245)    | 0.2421 | 0.2535 |
> | Static Transition Point (tau=937.8367)    | 0.2202 | 0.2452 |
> | Static Transition Point (tau=917.4490)    | 0.1972 | 0.2341 |
> | Batch-wise                                | 0.1976 | 0.2335 |
> | Per-sample (l_target=15)                  | 0.253  | 0.2576 |
> | Per-sample (l_target=10)                  | 0.2254 | 0.2479 |
> | Per-sample (l_target=8)                   | 0.1984 | 0.234  |
> | Per-sample (l_target=5)                   | 0.1712 | 0.2154 |
>
> > **W3 & Q7**: On the effectiveness of the proposed methods.
>
> - Performance of the proposed method
>
> To enable a more comprehensive evaluation, we varied the mitigation strength for both the baselines and our proposed methods using the settings from Table 1. Due to the character limit, we kindly refer you to our response to **Reviewer XBNj’s Question 2** for the corresponding performance table (related to Pareto front results).
>
> - On combining Per-sample with Prompt Embedding Adjustment
>
> The table below presents results for combining *Per-sample* with *Prompt Embedding Adjustment*, using the same settings as Table 1. In the combined approach (e.g., *l_target = 5* for Prompt Adjustment and *0.9* for Per-sample), prompt embedding is first adjusted until $||\tilde{\epsilon}_\theta(x_T, T, y)||_2 < 5$, followed by noise adjustment until it falls below 0.9.
>
> As shown, *Per-sample* alone achieves the best Pareto fronts, while *Prompt Embedding Adjustment* performs the worst. The combined method improves upon *Prompt Embedding Adjustment* but still falls short of *Per-sample*. This may be due to prompt modifications disrupting alignment, whereas *Per-sample* maintains alignment by keeping the prompt unchanged.

---

> ### Author Response · Authors · 2025-07-31
>
> |                                                                      | SSCD   | CLIP   |
> |----------------------------------------------------------------------|--------|--------|
> | Prompt Embedding Adjustment (l_target=4)                             | 0.1994 | 0.2545 |
> | Prompt Embedding Adjustment (l_target=5)                             | 0.2527 | 0.2644 |
> | Prompt Embedding Adjustment (l_target=6)                             | 0.3117 | 0.2733 |
> | Prompt Embedding Adjustment (l_target=5) + Per-sample (l_target=0.9) | 0.2163 | 0.2584 |
> | Prompt Embedding Adjustment (l_target=6) + Per-sample (l_target=0.9) | 0.2299 | 0.2617 |
> | Prompt Embedding Adjustment (l_target=7) + Per-sample (l_target=0.9) | 0.2352 | 0.2635 |
> | Per-sample (l_target=0.7)                                            | 0.1731 | 0.2505 |
> | Per-sample (l_target=0.9)                                            | 0.2265 | 0.2647 |
> | Per-sample (l_target=1.1)                                            | 0.2778 | 0.2743 |
>
> > **W4**: Pareto fronts across different hyperparameters.
>
> As shown in our response to Weakness 3, Per-sample, our proposed method, consistently achieves the best Pareto fronts across a range of hyperparameter settings.
>
> > **W5**: Suggestion to include FD_Dinov2.
>
> We agree that FD_Dinov2 offers a more semantically meaningful evaluation due to its stronger vision backbone. If resources permit, we plan to include it in the revised version. We appreciate the suggestion.
>
> > **Q1**: A quantitative study across a whole dataset.
>
> We evaluated transition point shifts across 5,000 samples generated from 500 memorized prompts, varying initial noise $x_T$ and adjustment strengths ($\tilde{\gamma} = 0.5$, $1.0$, $1.5$).
>
> For each $x_T$, we measured the change in transition point relative to the baseline ($\tilde{\gamma} = 0$). Positive values indicate earlier transitions.
>
> The table below shows average changes, all positive, confirming that initial noise adjustment consistently advances the transition point, with larger $\tilde{\gamma}$ producing greater shifts.
>
> |                     | gamma=1.5 & gamma=0.0 | gamma=1.0 & gamma=0.0 | gamma=0.5 & gamma=0.0 |
> |---------------------|-----------------------|-----------------------|-----------------------|
> | timestep difference | 35.1555               | 26.5776               | 20.6943               |
>
> > **Q3**: SNR based transition point analysis.
>
> Following prior work [3], we analyzed transition points using their corresponding log SNR values. Using a whole dataset, we measured how varying $\tilde{\gamma}$ ($0.5$, $1.0$, $1.5$) shifts the transition point relative to the baseline ($\tilde{\gamma} = 0$). Negative values indicate earlier transitions.
>
> As shown in the table, higher $\tilde{\gamma}$ consistently leads to earlier transition points, aligning with the trends observed in the timestep-based analysis.
>
> |                    | gamma=1.5 & gamma=0.0 | gamma=1.0 & gamma=0.0 | gamma=0.5 & gamma=0.0 |
> |--------------------|-----------------------|-----------------------|-----------------------|
> | log SNR difference | -1.7910               | -1.7146               | -0.2862               |
>
> > **Q4**: On the regions in the initial noise space.
>
> The region of the initial noise space where $x_T$ has a sufficiently small $||\tilde{\epsilon}_\theta(x_T, T, y)||_2$ value—small enough to eliminate the transition point—can be considered outside the attraction basin. Investigating how to directly sample initial noise from this region would be a promising direction for future research.
>
> > **Q5**: initial noise displacement.
>
> As shown in our response to Reviewer JH8S’s Question 1-2, JSD between 5,000 adjusted and unadjusted samples was only 0.0665, indicating minimal distribution shift. This suggests that the overall adjustment magnitude is likely small.
>
> > **Q6**: Computational Cost Analysis.
>
> Due to the character limit, we kindly ask you to refer to our response to Reviewer JH8S's Question 1-1 for further details.
>
> > **Q8**: On the high CFG.
>
> We followed prior works [2, 4, 5] in using a high CFG weight (e.g., 7.0–7.5). As noted in our response to Weakness 1, lower CFG weights improve FID but still produce high SSCD scores, indicating limited effectiveness in mitigating memorization.
>
> [1] Webster et al. "A reproducible extraction of training images from diffusion models". arXiv 2023.
>
> [2] Ren et al. "Unveiling and mitigating memorization in text-to-image diffusion models through cross attention". ECCV 2024.
>
> [3] Kingma et al. “Understanding Diffusion Objectives as the ELBO with Simple Data Augmentation”. NeurIPS 2023.
>
> [4] Jain et al. “Classifier-Free Guidance Inside the Attraction Basin May Cause Memorization”. CVPR 2025.
>
> [5] Hintersdorf et al. “Finding NeMo: Localizing Neurons Responsible For Memorization in Diffusion Models”. NeurIPS 2024.

---

> > ### Comment · Reviewer_tNu7 · 2025-08-02
> >
> > Thank you for the additional experiments, especially for the Pareto fronts on SD 2.0.
> >
> > **Performance vs baselines**
> >
> > I've plotted the Pareto fronts of your first table, and it seems like the Static Transition point performs equally to both per-sample and batch-wise adjustment. Could you comment on whether this is indeed a baseline method rather than an ablation of your method?
> >
> > **Cost Analysis**
> >
> > Thanks for providing it, it seems like the proposed method increases runtime by about 50%, which is to be expected.
> >
> > **Note to the AC**
> >
> > I cannot click on "acknowledge rebuttal" since the rebuttal is posted as official comment, so please consider the rebuttal as acknowledged with this comment.
> >
> > ---
> > For the moment, I will remain with my rating, but I am looking forward to your response on the first point and I am actively tracking the discussion with the other reviewers. I am still open to score increases, this is an interesting paper, the main question remains whether it performs strongly and robustly across models.

---

> > > ### Author Response · Authors · 2025-08-03
> > >
> > > We appreciate your thoughtful comments and would like to provide a response to your question.
> > >
> > > *Static Transition Point* is not part of our proposed method or its ablations; rather, it is one of the approaches introduced in prior work [1], and we included it as a baseline for evaluation on SDv2.0. Our reasons for selecting *Static Transition Point* as a baseline in this context, unlike in Table 1, are as follows:
> > >
> > > 1. According to Jain et al. [1], among the methods they proposed, *Dynamic Transition Point with Opposite Guidance* achieved the best performance on SDv1.4, while *Static Transition Point* performed best on fine-tuned SDv2.1. That is, *Dynamic Transition Point with Opposite Guidance* outperformed *Static Transition Point* on SDv1.4, but the reverse was true for fine-tuned SDv2.1.
> > > 2. In our own experiments on SDv2.0, *Dynamic Transition Point with Opposite Guidance* yielded significantly worse Pareto fronts compared to *Static Transition Point*.
> > >
> > > As you pointed out, *Static Transition Point* shows performance comparable to *Per-sample* and *Batch-wise* methods on SDv2.0. However, *Per-sample* not only achieves the best Pareto fronts on SDv2.0 but also consistently outperforms others on SDv1.4. We believe this demonstrates the effectiveness and robustness of our proposed method.
> > >
> > > We sincerely thank you once again for your thoughtful feedback and interest in our work.
> > >
> > > [1] Jain et al. "Classifier-Free Guidance inside the Attraction Basin May Cause Memorization". CVPR 2025.

---

> > > > ### Comment · Reviewer_tNu7 · 2025-08-04
> > > >
> > > > I thank the authors for the swift response. I have one last question left.
> > > >
> > > > **Static transition point baseline on SD 1.4**
> > > >
> > > > Given that the static transition point is a relatively strong baseline in your SD 2.0 experiment, could you try it on SD 1.4 to generate a similar Pareto front table across hyperparameters? If possible, just the same setup as above, with Pareto fronts for all methods, but if your compute is limited, even a sweep over the static transition point baseline would be good (with back of the envelope math, I think a single V100 should be able to compute 10*500 images in two days, bfloat-compatible accelerators being faster). It would be great if you could provide this to have a clearer picture of the performance landscape.

---

> > > > > ### Author Response · Authors · 2025-08-05
> > > > >
> > > > > We are very grateful for your constructive comments. In line with your suggestion, we conducted an evaluation of Static Transition Point on SDv1.4 using a range of hyperparameter values. Except for the method-specific hyperparameters, all other experimental settings followed those in Table 1 to ensure a fair comparison. The corresponding results are provided below.
> > > > >
> > > > > |                                                              | SSCD   | CLIP   |
> > > > > |--------------------------------------------------------------|--------|--------|
> > > > > | Static Transition Point (tau=780)                            | 0.2829 | 0.2600 |
> > > > > | Static Transition Point (tau=760)                            | 0.2534 | 0.2523 |
> > > > > | Static Transition Point (tau=740)                            | 0.2244 | 0.2441 |
> > > > > | Static Transition Point (tau=720)                            | 0.1975 | 0.2360 |
> > > > > | Static Transition Point (tau=700)                            | 0.1762 | 0.2274 |
> > > > > | Opposite Guidance with Dynamic Transition Point (og_scale=1) | 0.2396 | 0.2409 |
> > > > > | Opposite Guidance with Dynamic Transition Point (og_scale=3) | 0.2136 | 0.2403 |
> > > > > | Opposite Guidance with Dynamic Transition Point (og_scale=5) | 0.1908 | 0.2315 |
> > > > > | Opposite Guidance with Dynamic Transition Point (og_scale=7) | 0.1711 | 0.2202 |
> > > > > | Opposite Guidance with Dynamic Transition Point (og_scale=9) | 0.1247 | 0.2003 |
> > > > > | Batch-wise                                                   | 0.2642 | 0.2603 |
> > > > > | Per-sample (l_target=0.7)                                    | 0.1731 | 0.2505 |
> > > > > | Per-sample (l_target=0.9)                                    | 0.2265 | 0.2647 |
> > > > > | Per-sample (l_target=1.1)                                    | 0.2770 | 0.2743 |
> > > > > | Per-sample (l_target=1.3)                                    | 0.3220 | 0.2810 |
> > > > > | Per-sample (l_target=1.5)                                    | 0.3593 | 0.2850 |
> > > > >
> > > > > As the results indicate, *Static Transition Point* and *Dynamic Transition Point with Opposite Guidance* exhibit similar Pareto fronts, with *Static Transition Point* performing slightly better. However, both are outperformed by our proposed methods, *Per-sample* and *Batch-wise*, which achieve notably stronger Pareto fronts. This confirms that our methods consistently provide superior mitigation performance compared to existing baselines on SDv1.4.
> > > > >
> > > > > (For additional baseline results not included in the table above, we kindly refer you to our response to **Reviewer XBNj’s Question 2**, which presents a table based on the same experimental settings used here.)
> > > > >
> > > > > Upon re-examining prior work [1], we found no direct comparison between *Static Transition Point* and *Dynamic Transition Point with Opposite Guidance*. We apologize for any confusion this may have caused. Nonetheless, our empirical results on both SDv1.4 and SDv2.0 highlight not only the effectiveness but also the robustness of our proposed approach.
> > > > >
> > > > > We sincerely appreciate your continued thoughtful feedback.
> > > > >
> > > > > [1] Jain et al. "Classifier-Free Guidance inside the Attraction Basin May Cause Memorization". CVPR 2025.

---

> > > > > > ### Comment · Reviewer_tNu7 · 2025-08-09
> > > > > >
> > > > > > Thank you for the experiment! I take it that Static Transition Point performs worse than the proposed method on SD 1.4 and equally on SD 2.0. This was my last concern, and I thank you for the productive discussion period. I will read through the discussions with the other reviewers and update my score accordingly.

---

> > > > > > > ### Author Response · Authors · 2025-08-09
> > > > > > >
> > > > > > > We are glad to hear that your last concern has been fully resolved. Thank you for dedicating your time to actively participating in the discussion and for showing interest in our paper.

---

### Official Review · Reviewer_ye9A · 2025-07-01

**Clarity:** 3
**Significance:** 2
**Originality:** 3
**Rating:** 5
**Confidence:** 5

**Summary:**

Previous research has found an attraction basin that leads to memorization in T2I Diffusion models. Deferring the use of classifier-free guidance helps prevent memorization. However, if the transition point of escaping the attraction basin is in later timesteps, the alignment of the image with the given prompt is poor. Therefore, this paper presents the finding that with different initial noise initializations, the transition point can be in earlier or later timesteps.
As a mitigation, two methods are proposed: searching for the optimal initial noise to shift the transition point to earlier timesteps, effectively mitigating memorization.

**Questions:**

- Q1: Why is the FID score so high? On what dataset was the FID score calculated? Is the FID score calculated on the memorized samples?
- Q2: Why is the FID score improving when performing mitigation?

**Ethical Concerns:**

["NO or VERY MINOR ethics concerns only"]

**Final Justification:**

My concerns have been appropriately addressed, which is why I raise my score to "accept".

**Limitations:**

Yes

**Paper Formatting Concerns:**

No concerns

**Quality:**

3

**Strengths And Weaknesses:**

**Strengths:**
- The paper presents the critical insight that the choice of the initial noise highly influences the degree of memorization
- The paper is well written
- The paper is easy to read and easy to understand

**Weaknesses:**
- In the batch-wise mitigation, multiple additional time steps need to be computed
- Finding the initial noise is quite computationally intensive because optimization needs to be done
- It is a bit confusing that in the experiments 50 time steps are used, but in the graphs in Figure 3 1000 time steps are shown
- Comparison to the inference time mitigation of Hintersdorf et al. is missing

---

> ### Author Rebuttal · Authors · 2025-07-31
>
> We are very grateful for your constructive comments. We have provided answers to each comment. Please let us know if you need any clarification or have additional questions.
>
> > **W1 & W2**: Computational Cost Analysis.
>
> To ensure a fair comparison, we measured GPU memory usage and inference time per image. As shown in the table, Batch-wise incurs minimal overhead, while Per-sample requires slightly more resources.
>
> Nonetheless, as addressed in our responses to Reviewer tNu7’s Weaknesses 2 and 4, Per-sample consistently achieves the best Pareto fronts (SSCD vs. CLIP) across SDv1.4 and SDv2.0 compared to other baselines.
>
> These findings suggest that Per-sample offers superior memorization mitigation performance and robustness across different models. As discussed in Section 6, effective mitigation of memorization plays a crucial role in enhancing privacy and intellectual property protection while significantly reducing potential risks. Considering this, we believe that the additional cost of the Per-sample method is a reasonable trade-off.
>
> |                                                 | gpu memory usage (MB) | inference time (sec) |
> |-------------------------------------------------|-----------------------|----------------------|
> | No Mitigation                                   | 3564.94               | 1.85                 |
> | Adding 4 Random Tokens                          | 3573.57               | 1.85                 |
> | Prompt Embedding Adjustment (l_target=3)        | 8115.53               | 1.98                 |
> | Cross-attention Scaling                         | 3170.38               | 1.93                 |
> | Opposite Guidance with Dynamic Transition Point | 3573.57               | 1.87                 |
> | Ours (Batch-wise)                               | 3573.57               | 2.01                 |
> | Ours (Per-sample)                               | 9701.62               | 2.8                  |
>
> > **W3**: Used timesteps in the experiments.
>
> Following the experimental setup of prior work [1], we report our results in Figure 3. Using 50 inference steps for denoising results in a step size of 20, meaning the timesteps used are [981, 961, 941, …, 1].
>
> > **W4**: On the missing inference time mitigation baseline (Hintersdorf et al.)
>
> Our proposed method is an inference-time mitigation approach that does not require access to training data. Therefore, as described in Section 4.1, we selected baseline methods that also operate at inference time and do not rely on training data, to ensure a fair comparison.
>
> To the best of our knowledge, NeMo [2] requires access to training data to compute activation statistics on unmemorized samples.
>
> > **Q1**: On the calculation of FID score.
>
> We computed the FID using 10000 randomly sampled images from the LAION dataset and 5000 generated images produced from memorized prompts. We suspect that the number of generated and reference images, along with the exclusive use of memorized prompts for generation, may have influenced the scale of the FID score.
>
> > **Q2**: The reason why mitigation methods improve FID score.
>
> Without mitigation, diffusion models tend to generate identical or highly similar images for memorized prompts, which significantly reduces diversity. In contrast, applying mitigation alleviates this issue, increases diversity, and is likely to improve the FID score as a result.
>
> The table below extends Table 1 by including LPIPS (a diversity metric) and ImageReward (a metric for quality and prompt alignment), where higher values are better for both. As shown, applying mitigation leads to higher LPIPS scores, confirming that diversity indeed improves.
>
> |                                                 | SSCD   | CLIP Score | FID      | LPIPS  | ImageReward |
> |-------------------------------------------------|--------|------------|----------|--------|-------------|
> | No Mitigation                                   | 0.5439 | 0.3012     | 160.4486 | 0.4835 | -0.014      |
> | Adding 1 Random Token                           | 0.4794 | 0.2903     | 146.8346 | 0.5471 | -0.1013     |
> | Adding 4 Random Tokens                          | 0.3402 | 0.2608     | 127.2910 | 0.6671 | -0.3532     |
> | Prompt Embedding Adjustment (l_target=3)        | 0.1325 | 0.2372     | 97.5206  | 0.7522 | -0.6466     |
> | Prompt Embedding Adjustment (l_target=5)        | 0.2527 | 0.2644     | 108.8194 | 0.7003 | -0.3338     |
> | Cross-attention Scaling                         | 0.2779 | 0.2661     | 117.6667 | 0.7194 | -0.2115     |
> | Opposite Guidance with Dynamic Transition Point | 0.2396 | 0.2409     | 94.2193  | 0.7502 | -0.6639     |
> | Ours (Batch-wise)                               | 0.2642 | 0.2603     | 105.5606 | 0.7384 | -0.4527     |
> | Ours (Per-sample)                               | 0.2265 | 0.2647     | 111.7207 | 0.7615 | -0.3522     |
>
>
> [1] Jain et al. "Classifier-Free Guidance inside the Attraction Basin May Cause Memorization". CVPR 2025.
>
> [2] Hintersdorf et al. "Finding nemo: Localizing neurons responsible for memorization in diffusion models". NeurIPS 2024.

---

> > ### Comment · Reviewer_ye9A · 2025-08-04
> >
> > Thank you for your detailed answer.
> > Most of my concerns have been appropriately addressed.
> >
> > However, one question remains. In the second table you have posted the "Opposite Guidance with Dynamic Transition Point" seems to be better in every metric.
> > This brings up the following question: What is the benefit of your proposed solution in comparison to the dynamic transition point method?

---

> > > ### Author Response · Authors · 2025-08-05
> > >
> > > We sincerely appreciate your interest in our paper and your thoughtful engagement in the discussion. We would like to address the question you raised.
> > >
> > > First, regarding the interpretation of the evaluation metrics: lower values indicate better performance for SSCD and FID, while higher values are preferred for CLIP score, LPIPS, and ImageReward.
> > >
> > > As noted in Section 4.1, SSCD evaluates object-level similarity between a generated image and its nearest neighbor in the training set, and has therefore been widely adopted as a key metric for assessing the degree of memorization in diffusion models. However, SSCD alone may not provide a complete picture, as it is important that memorization mitigation does not come at the expense of the model’s original generative capabilities.
> > >
> > > To this end, additional metrics such as CLIP score and FID are often used to capture aspects of image-text alignment and image quality, respectively. In fact, many prior methods have shown that efforts to reduce memorization often lead to noticeable declines in image-text alignment. Consequently, analyzing how SSCD and CLIP scores change across various hyperparameter settings—and identifying methods that yield favorable Pareto fronts—has become a common and informative strategy for performance evaluation.
> > >
> > > Indeed, several previous works [1, 2] have adopted this approach, reporting trade-offs between SSCD and CLIP score and using the resulting Pareto fronts as indicators of overall method effectiveness.
> > >
> > > Consistent with this perspective, we also evaluated SSCD and CLIP scores across a range of hyperparameter settings and compared the resulting Pareto fronts between our proposed methods and the baselines. For a detailed summary of the results, we kindly refer you to the table included in our response to **Reviewer XBNj’s Question 2**. This table covers performance across various hyperparameters, including those used in Table 1 of the paper, and thus encompasses the data referenced in what you described as "the second table."
> > >
> > > Based on this table, plotting SSCD vs. CLIP curves reveals that *Per-sample* yields more favorable Pareto fronts than all other baselines, including *Opposite Guidance with Dynamic Transition Point*. In other words, *Per-sample* consistently exhibits the most favorable trade-off, indicating the strongest mitigation performance.
> > >
> > > Additionally, in terms of ImageReward, *Opposite Guidance with Dynamic Transition Point* generally shows lower scores compared to *Per-sample* and *Batch-wise*, suggesting that it generates images with lower quality and weaker prompt alignment.
> > >
> > > In summary, our proposed method demonstrates superior Pareto fronts (in terms of SSCD and CLIP) compared to *Opposite Guidance with Dynamic Transition Point*, highlighting its strong mitigation performance.
> > >
> > > [1] Wen et al. “Detecting, Explaining, and Mitigating Memorization in Diffusion Models”. ICLR 2024.
> > >
> > > [2] Ren et al. “Unveiling and mitigating memorization in text-to-image diffusion models through cross attention”. ECCV 2024.

---

> > > > ### Comment · Reviewer_ye9A · 2025-08-05
> > > >
> > > > Thank you for clarifying. My concerns have been addressed, which is why I raise my score.

---

> > > > > ### Author Response · Authors · 2025-08-05
> > > > >
> > > > > We’re glad to hear that our response addressed all of your concerns. We sincerely appreciate the time you took to review our response and your continued interest in our work. We are also grateful for your decision to raise the score.

---

### Official Review · Reviewer_XBNj · 2025-07-02

**Clarity:** 3
**Significance:** 3
**Originality:** 3
**Rating:** 4
**Confidence:** 5

**Summary:**

The task that this paper tackles is the memorization issue in text-to-image diffusion models, which represent the fact that these models often memorize and replicate training data, leading to privacy and copyright concerns. This paper builds on a recent work by Jain et al., where the baseline method views this problem in the lens of the attraction basin, which refers to a region where applying classifier-free guidance (CFG) can cause memorization during denoising. Also, the baseline paper identifies a phenomenon that the earlier a generation can escape from such region, the better alignment can be preserved during the memorization mitigation. Based on these insights, this paper proposes a new mitigation strategy that adjust the initial noise by search initializations that are more easily to escape from the region. Two variants of noise adjustment methods are propose: batch-wise and per-sample. Experiments are conducted on Stable Diffusion v1.4 architecture.

**Questions:**

- It is recommended to also present the comparisons of performance with and without adjustment for the per-sample strategy in addition to the current Figure 4.
- The effectiveness of the proposed methods are not convincing.
- A more comprehensive comparison over different privacy levels is very important for effectively comparing performances across different mitigation strategies.
- Can you explain the experimental settings you used for producing Figure 4?

**Ethical Concerns:**

["NO or VERY MINOR ethics concerns only"]

**Final Justification:**

The rebuttal has effectively addressed my major concerns regarding the experimental performance and insufficient comparisons. Therefore, I have increased my final rating from 3 to 4 and the significance rating from 1 to 3. I recommend that the authors include the revised results in a Figure format for better presentation of a more comprehensive result (which is understandable and perfectly fine to provide tables in the rebuttal due to the constraint that Figures are not supported in this rebuttal).

**Limitations:**

Yes.

**Paper Formatting Concerns:**

No formatting concerns.

**Quality:**

2

**Strengths And Weaknesses:**

**Strengths**:

- The writing of this paper is clear and structured. Visuals are well-employed to facilitate the discussions. Thus, I would evaluate its clarity to be good.
- The motivation of the addressed problem is well-justified, making it convincing for the readers that mitigating memorization in text-to-image diffusion models is a practical challenge that needs to be addressed.
- The proposed mitigation strategy is developed from a novel perspective (i.e., adjusting the initial noise) from existing methods that tackle this problem by modifying the denoising process during inference.

**Weaknesses**:

- The baseline comparisons are quite insufficient as the paper currently only compares with one or two privacy levels for each compared method. For example, the cross-attention scaling method is only shown in one privacy level, and the random token and prompt embedding adjustment methods are only shown in two privacy levels. However, it is essential to have much more results compared under different privacy levels to sufficiently compare how much alignment (CLIP score) or fidelity (FID) is sacrificed, so that the effectiveness can be demonstrated.
- The effectiveness of the proposed method appears suboptimal compared to the baselines. For example, the batch-wise strategy shows inferior SSCD and CLIP scores relative to the baseline from Wen et al. Additionally, all three metrics (SSCD, CLIP score, and FID) indicate that the proposed method underperforms compared to the baseline from Jain et al. The per-sample strategy also frequently yields worse results than the baseline on one or two of these metrics. Combined with the earlier concern about the very limited scope of the current evaluation, which means the proposed method already struggles in this narrow performance snapshot, let alone in a broader comparison across different privacy levels. Therefore, I am concerned about the paper’s practical significance.
- The comparisons of performance with and without adjustment should be conducted for both batch-wise and per-sample strategies. However, only the batch-wise strategy’s results are shown in Figure 4. The authors also acknowledged that the performance of batch-wise strategy is quite sensitive to the choice of hyperparameter values, while the effectiveness is mainly demonstrated in the per-sample strategy. This makes it more important to present the result for per-sample strategy.
- The settings of Figure 4 need to be elaborated in more details: are the settings align with the ones used in creating Table 1 that both follow the setup in section 4.1? If yes, does the “no adjustment” version correspond to one of the baseline versions?

---

> ### Author Rebuttal · Authors · 2025-07-31
>
> We thank the reviewer for the constructive comments. We have provided answers to each comment. Please let us know if you need any clarification or have additional questions.
>
> > **Q1**: Comparisons of performance with and without adjustment for the Per-sample strategy.
>
> As described in Section 3.2.2, the *Per-sample* strategy involves adjusting the initial noise $x_T$ and applying CFG from the beginning of the sampling process. Without the initial noise adjustment, hence, the *Per-sample* strategy becomes equivalent to the *No Mitigation.*
>
> Therefore, the performance comparison between *No Mitigation* and *Per-sample* reported in Table 1 allows us to assess the effect of applying the initial noise adjustment. Additionally, Appendix D.3 presents the performance of the *Per-sample* strategy under different adjustment strengths. The comprehensive results are summarized in the table below, demonstrating that initial noise adjustment effectively mitigates memorization.
>
> |                           | SSCD   | CLIP Score |
> |---------------------------|--------|------------|
> | No Mitigation             | 0.5439 | 0.3012     |
> | Per-sample (l_target=1.5) | 0.3593 | 0.2850     |
> | Per-sample (l_target=1.3) | 0.3220 | 0.2810     |
> | Per-sample (l_target=1.1) | 0.2778 | 0.2743     |
> | Per-sample (l_target=0.9) | 0.2265 | 0.2647     |
> | Per-sample (l_target=0.7) | 0.1731 | 0.2505     |
>
> > **Q2**: The effectiveness of the proposed methods.
>
> To conduct a more comprehensive performance evaluation, we varied the mitigation strength for both the baselines and our proposed methods. The results are summarized in the table below, which also includes evaluations using LPIPS (diversity) and ImageReward (quality and prompt alignment). All other experimental settings are consistent with those in Table 1.
>
> Our main finding is that adjusting the initial noise $x_T$ to minimize $||\tilde{\epsilon}_\theta(x_T, T, y)||_2$ leads to an earlier transition point and improves mitigation performance. Based on this, we proposed two strategies—*Per-sample* and *Batch-wise*—which share the same core idea.
>
> *Batch-wise* involves three hyperparameters ($\tilde{\gamma}$, $\rho$, and $M$), while *Per-sample* uses one (l_target). Since mitigation strength is closely related to the value of $||\tilde{\epsilon}_\theta(x_T, T, y)||_2$, we report the performance of Per-sample across different l_target values.
>
> Using the data in the table, one can plot SSCD vs. CLIP curves and observe that *Per-sample* achieves the best Pareto fronts. *Batch-wise* also shows strong performance, comparable to *Cross-attention Scaling*. These results demonstrate the effectiveness of our proposed methods.
>
> While we are unable to include figures in this rebuttal, we will provide graphical results in the revised version of the paper.
>
> |                                                              | SSCD   | CLIP   | FID      | LPIPS  | ImageReward |
> |--------------------------------------------------------------|--------|--------|----------|--------|-------------|
> | No Mitigation                                                | 0.5439 | 0.3012 | 160.4486 | 0.4835 | -0.014      |
> | Adding 1 Random Token                                        | 0.4794 | 0.2903 | 146.8346 | 0.5471 | -0.1013     |
> | Adding 2 Random Tokens                                       | 0.4171 | 0.2778 | 132.5187 | 0.5714 | -0.1581     |
> | Adding 4 Random Tokens                                       | 0.3402 | 0.2608 | 127.291  | 0.6671 | -0.3532     |
> | Adding 6 Random Tokens                                       | 0.2728 | 0.2405 | 110.9416 | 0.679  | -0.5496     |
> | Adding 8 Random Tokens                                       | 0.2147 | 0.2240 | 104.5567 | 0.7098 | -0.7387     |
> | Prompt Embedding Adjustment (l_target=3)                     | 0.1325 | 0.2372 | 97.5206  | 0.7522 | -0.6466     |
> | Prompt Embedding Adjustment (l_target=4)                     | 0.1994 | 0.2545 | 104.7461 | 0.723  | -0.443      |
> | Prompt Embedding Adjustment (l_target=5)                     | 0.2527 | 0.2644 | 108.8194 | 0.7003 | -0.3338     |
> | Prompt Embedding Adjustment (l_target=6)                     | 0.3117 | 0.2733 | 116.4698 | 0.668  | -0.251      |
> | Prompt Embedding Adjustment (l_target=7)                     | 0.3586 | 0.2811 | 123.8324 | 0.6355 | -0.1835     |
> | Cross-attention Scaling (c=1.15)                             | 0.3094 | 0.2732 | 121.5525 | 0.6983 | -0.1236     |
> | Cross-attention Scaling (c=1.2)                              | 0.2935 | 0.2696 | 119.1731 | 0.7095 | -0.1666     |
> | Cross-attention Scaling (c=1.25)                             | 0.2779 | 0.2661 | 117.6667 | 0.7194 | -0.2115     |
> | Cross-attention Scaling (c=1.3)                              | 0.2662 | 0.2632 | 117.2156 | 0.7275 | -0.2506     |
> | Cross-attention Scaling (c=1.35)                             | 0.2566 | 0.2609 | 117.0359 | 0.7329 | -0.2803     |
> | Opposite Guidance with Dynamic Transition Point (og_scale=1) | 0.2396 | 0.2409 | 94.2193  | 0.7502 | -0.6639     |
> | Opposite Guidance with Dynamic Transition Point (og_scale=3) | 0.2136 | 0.2403 | 93.8351  | 0.772  | -0.6772     |
> | Opposite Guidance with Dynamic Transition Point (og_scale=5) | 0.1908 | 0.2315 | 90.0785  | 0.7897 | -0.766      |
> | Opposite Guidance with Dynamic Transition Point (og_scale=7) | 0.1711 | 0.2202 | 87.6463  | 0.8367 | -0.8884     |
> | Opposite Guidance with Dynamic Transition Point (og_scale=9) | 0.1247 | 0.2003 | 84.7859  | 0.8393 | -1.1194     |
> | Batch-wise                                                   | 0.2642 | 0.2603 | 105.5606 | 0.7384 | -0.4527     |
> | Per-sample (l_target=0.7)                                    | 0.1731 | 0.2505 | 106.06   | 0.8028 | -0.5552     |
> | Per-sample (l_target=0.9)                                    | 0.2265 | 0.2647 | 111.7207 | 0.7615 | -0.3522     |
> | Per-sample (l_target=1.1)                                    | 0.2770 | 0.2743 | 117.8896 | 0.7174 | -0.2556     |
> | Per-sample (l_target=1.3)                                    | 0.322  | 0.281  | 124.5573 | 0.6796 | -0.1847     |
> | Per-sample (l_target=1.5)                                    | 0.3593 | 0.285  | 128.595  | 0.6473 | -0.1414     |
>
> > **Q3**: A more comprehensive comparison over different privacy levels.
>
> Through the above response, we provide a comprehensive comparison across different privacy levels. The fact that Per-sample achieves the best Pareto fronts further demonstrates the effectiveness of our proposed method.
>
> > **Q4**: The experimental settings used in Figure 4.
>
> The experimental setup in Figure 4 follows the same configuration as in Table 1; however, the *No adjustment* shown in Figure 4 does not correspond to any of the baselines reported in Table 1. Before explaining this further, we would like to clarify the purpose of Figure 4.
>
> Our goal was to examine whether initial noise adjustment meaningfully affects mitigation performance. For this, we used the *Batch-wise* strategy. The mitigation performance of *Batch-wise* is primarily influenced by two factors: (1) whether initial noise adjustment is applied, and (2) the starting timestep for applying CFG (denoted as $\tau$).
>
> We observed that increasing $\tau$—that is, applying CFG earlier—tends to improve both SSCD and CLIP scores, indicating a trade-off between the two. Importantly, this trend holds consistently regardless of whether initial noise adjustment is used. This is likely because the point at which a sample exits the attraction basin varies depending on the seed (i.e., $x_T$) and the prompt; thus, applying CFG too early (i.e., at higher $\tau$) increases the chance of generating memorized images.
>
> For this reason, we varied the $\tau$ values under both settings—with and without initial noise adjustment—and reported the results in Figure 4, labeled as *With adjustment* and *No adjustment*, respectively. As shown, initial noise adjustment yields a more favorable trade-off between SSCD and CLIP scores compared to the *No adjustment*. This demonstrates that initial noise adjustment indeed enhances memorization mitigation performance.
>
> (Note: In Figure 4, the rightmost data point on the *No adjustment* curve corresponds to $\tau = 780$, and the leftmost to $\tau = 720$. Similarly, the rightmost and leftmost data points on the *With adjustment* curve correspond to $\tau = 900$ and $\tau = 840$, respectively.)

---

> > ### Comment · Reviewer_XBNj · 2025-08-05
> >
> > Thanks for the authors' rebuttal, which has effectively addressed my major concerns regarding the experimental performance and insufficient comparisons. I'll increase my ratings (final rating and significance rating) accordingly.

---

> > > ### Author Response · Authors · 2025-08-05
> > >
> > > Thank you for taking the time to read our response. We truly appreciate your thoughtful evaluation of our work and your decision to increase the score. We will carefully reflect on the points raised during the discussion and incorporate them into the revised version of the paper.

---

### Official Review · Reviewer_JH8S · 2025-07-03

**Clarity:** 3
**Significance:** 3
**Originality:** 4
**Rating:** 4
**Confidence:** 4

**Summary:**

This paper introduces a novel, training-free approach to mitigate memorization in text-to-image diffusion models by focusing on the initial noise sample, $x_T$. The work is grounded in the "attraction basin" theory, which suggests memorization occurs when classifier-free guidance (CFG) steers the generation process towards a training example. The authors' core insight is that the choice of $x_T$ significantly impacts when the denoising trajectory escapes this basin. They empirically demonstrate a correlation between a lower initial conditional guidance magnitude and an earlier, more desirable escape time. Based on this, they propose two inference-time strategies to find such favorable initializations: a fast "Batch-wise" method that adjusts a batch of samples collectively using a few forward passes, and a more intensive "Per-sample" method that uses iterative backpropagation to optimize each $x_T$ individually until a target loss is met. Experiments on Stable Diffusion v1.4 show that these methods significantly reduce memorization while preserving strong image-text alignment.

**Questions:**

- **On the Per-sample Optimization Loop:** Could the authors clarify the details of the per-sample optimization. Specifically:
	- (1) What is the average runtime or number of iterations required for the optimization loop to converge?
	- (2) Could you justify the choice of AdamW and explain the effect of its weight decay? Simple gradient descent seems like a more natural choice due to the optimization problem being non-stochastic. The effect of weight decay on the noise latents is unclear. How does it affect the performance of the method? How does it compare to standard Adam or gradient descent in this context?
	- (3) The method does not regularize the latent $x_T$ to keep it in a high likelihood region of the initial Gaussian distribution, as previous methods have proposed [1, 2]. Have the authors observed any "reward hacking" behavior where the optimization pushes the latent to an out-of-distribution state to minimize the objective? For instance, what happens to the generated images if the optimization loop is allowed to run for a significantly longer duration? Is there a reason why this type of regularization is not needed here?
- **On Decomposing Evaluation into Quality and Diversity:** The evaluation relies on global metrics like FID and CLIP score, which can make it difficult to disentangle the specific effects of the proposed methods. FID is a composite metric reflecting both image fidelity and diversity, the lack of clear correlation with CLIP score in Table 1 suggests these entangled aspects might be obscuring the true trade-offs.
    - To provide a clearer picture, have the authors considered evaluating these two axes separately? For **diversity**, one could measure the average pairwise similarity (e.g., using LPIPS or DINO like in Dreambooth [3]) between images generated from the same prompt, which would directly quantify if the methods reduce variance. For **quality and prompt alignment**, a more direct reward model like ImageReward could offer a more nuanced signal than CLIP or FID alone.
- **On Synergy with Other Methods:** Have the authors considered if this memorization mitigation technique could be combined with other noise optimization approaches that are designed to improve visual quality or prompt adherence?
- **On the Generality of the Framework:** The proposed method is fundamentally tied to CFG. For a purely unconditional model, one might hypothesize that memorization corresponds to anomalies like an unusually high unconditional score norm ($||ϵ_θ(x_t, t)||_2$). Do the authors believe their core principle, adjusting $x_T$ to minimize a proxy signal, could be adapted to the unconditional case using this or a similar intrinsic signal?

[1] Ben-Hanmu et al. "D-Flow: Differentiating through Flows for Controlled Generation". ICML 2024.

[2] Eyring et al. "ReNO: Enhancing One-step Text-to-Image Models through Reward-based Noise Optimization". NeurIPS 2024.

[3] Ruiz et al. "DreamBooth: Fine Tuning Text-to-Image Diffusion Models for Subject-Driven Generation". CVPR 2023.

**Ethical Concerns:**

["NO or VERY MINOR ethics concerns only"]

**Final Justification:**

After careful consideration of the authors' rebuttal and our subsequent discussion, I maintain my score of 4: Borderline accept.

**Issues Successfully Resolved:**

- Computational cost analysis was provided with detailed GPU memory and timing comparisons
- The choice of AdamW optimizer was justified with comparative experiments showing superior performance over SGD and Adam
- Evaluation was enhanced with LPIPS and ImageReward metrics, providing better separation of quality and diversity effects
- LPIPS computation methodology was clarified, confirming higher scores indicate greater diversity as intended
- SGD performance differences were explained as resulting from learning rate tuning rather than fundamental algorithmic issues

**Areas for Improvement:**

- Theoretical presentation: The discussion revealed that weight decay serves as implicit regularization to keep noise in-distribution, which is a fundamental aspect of the method's theoretical soundness that was not included in the original paper
- Methodological documentation: Key design choices such as the regularization strategy would benefit from more comprehensive explanation in the main text
- Implementation details: The method's sensitivity to optimizer choice and hyperparameter tuning suggests some aspects could be better documented for reproducibility

**Overall Assessment:**

The paper presents a novel and valuable perspective on memorization in diffusion models with practical mitigation strategies. The empirical results are convincing and the core insights about initial noise impact are important. The authors demonstrated good responsiveness to feedback and committed to including important theoretical details in the revision. The technical contributions and practical value support acceptance, though some methodological aspects could be better documented in the original submission.

**Limitations:**

The proposed framework is specifically designed for diffusion models that use Classifier-Free Guidance, which makes it not directly applicable to unconditional diffusion models.

**Quality:**

3

**Strengths And Weaknesses:**

**Strengths**:
- **A Novel Perspective on the Role of Initial Noise (Originality/Quality):** The paper's primary contribution is its novel perspective on the impact of the initial noise sample on memorization. The empirical observation that different initializations lead to varying escape times from the "attraction basin" is an important insight in itself, separate from the mitigation methods built upon it.
- **Practical and Effective Mitigation Methods (Originality/Signifcance):** Building on this core insight, the paper proposes two effective methods that cater to different needs. The "Batch-wise" approach is computationally cheap and highly practical, while the "Per-sample" method, though more intensive, achieves stronger results, providing a valuable trade-off for users. The per-sample method is elegantly formulated by defining an objective based on a single model evaluation at the initial timestep formulating an efficient test-time optimization problem.

**Weaknesses:**
- **Missing Computational Cost Analysis:** A significant weakness is the lack of a comparative computational cost analysis (e.g., in wall-clock time, NFEs, or GPU memory (for per-sample)) across all evaluated methods. This omission makes it hard to assess a performance vs. cost trade-off, e.g. also for the proposed per-sample vs per-batch methods.
- **Further Justification of Methodological Choices:** The per-sample optimization strategy would benefit from further justification. The choice of the AdamW optimizer is not clearly explained, particularly the potential effect of weight decay on a noise latent. Furthermore, the absence of a regularization term to keep the optimized noise within the initial Gaussian distribution raises questions about the potential for out-of-distribution "reward hacking."
- **Opportunity to Enhance Evaluation Metrics:** The evaluation could be enhanced by decoupling different aspects of generation quality. Relying primarily on global metrics like FID can make it difficult to pinpoint the specific effects of the methods on distinct axes like image fidelity versus sample diversity. This presents an opportunity to provide a more granular understanding of the methods' impact.

---

> ### Author Rebuttal · Authors · 2025-07-31
>
> Thank you for your insightful comments. We've carefully considered them and provided responses. Please let us know if you need any clarification or have additional questions.
>
> > **Q1-1**: Computational Cost Analysis.
>
> To ensure a fair comparison, we measured GPU memory usage and inference time per image. As shown in the table, Batch-wise incurs minimal overhead, while Per-sample requires slightly more resources.
>
> Nonetheless, as addressed in our responses to Reviewer tNu7’s Weaknesses 2 and 4, Per-sample consistently achieves the best Pareto fronts (SSCD vs. CLIP) across SDv1.4 and SDv2.0 compared to other baselines.
>
> These findings suggest that Per-sample offers superior memorization mitigation performance and robustness across different models. As discussed in Section 6, effective mitigation of memorization plays a crucial role in enhancing privacy and intellectual property protection while significantly reducing potential risks. Considering this, we believe that the additional cost of the Per-sample method is a reasonable trade-off.
>
> |                                                 | gpu memory usage (MB) | inference time (sec) |
> |-------------------------------------------------|-----------------------|----------------------|
> | No Mitigation                                   | 3564.94               | 1.85                 |
> | Adding 4 Random Tokens                          | 3573.57               | 1.85                 |
> | Prompt Embedding Adjustment (l_target=3)        | 8115.53               | 1.98                 |
> | Cross-attention Scaling                         | 3170.38               | 1.93                 |
> | Opposite Guidance with Dynamic Transition Point | 3573.57               | 1.87                 |
> | Ours (Batch-wise)                               | 3573.57               | 2.01                 |
> | Ours (Per-sample)                               | 9701.62               | 2.8                  |
>
> > **Q1-2**: The choice of AdamW and the effect of weight decay.
> - On the choice of AdamW
>
> As shown in Section 3.1.2, we observed that initial noise with a smaller conditional guidance leads to an earlier transition point, which in turn improves memorization mitigation performance. Therefore, our goal is to effectively reduce the norm $||\tilde{\epsilon}_\theta(x_T, T, y)||_2$.
>
> We experimented with minimizing this norm using both SGD and AdamW. The results show that AdamW reduces $||\tilde{\epsilon}_\theta(x_T, T, y)||_2$ more effectively than SGD. This leads to a substantial improvement in mitigation performance, as demonstrated in the table below. As shown in the table, AdamW achieves the best mitigation performance, and we therefore adopt it as our optimizer.
>
> |                    | SSCD   | CLIP score | FID      |
> |--------------------|--------|------------|----------|
> | AdamW              | 0.2265 | 0.2647     | 111.7207 |
> | Adam               | 0.2265 | 0.2647     | 111.7207 |
> | SGD (momentum=0.9) | 0.4209 | 0.294      | 131.2913 |
> | SGD (momentum=0)   | 0.5326 | 0.301      | 151.0111 |
>
> - The effect of weight decay
>
> We believe that weight decay plays a partial role in preventing adjusted initial samples from entering an out-of-distribution (OOD) state. This belief is grounded in the following reasoning. According to prior works [1, 2], the L2 norm of the initial noise $x_T \sim \mathcal{N}(0, I)$ follows a chi distribution:
>
> $||x_T|| = \sqrt{\sum_{i=1}^d {x_T^i}^2} \sim \chi^d = ||x_T||^{d-1}e^{-||x_T||^2/2}/(2^{d/2-1}\Gamma({d \over 2}))$,
>
> where $d$ is the dimensionality of $x_T$ and $\Gamma(\cdot)$ is the Gamma function. Based on this, to prevent the adjusted initial noise from deviating into an OOD state, one could consider minimizing the objective $||\tilde{\epsilon}_\theta(x_T, T, y)||_2 -w\log{p(||x_T||)}$, where $p$ is the chi distribution and $w$ is a weight (hyperparameter). Since $p$ is the chi distribution, this objective can be rewritten as:
>
> $||\tilde{\epsilon}_\theta(x_T, T, y)||_2 + c - {w (d-1)\log{||x_T||} + {w{||x_T||^2 \over 2}}}$,
>
> where $c$ is a constant independent of $x_T$. We argue that AdamW's weight decay implicitly minimizes the final term, ${||x_T||^2 \over 2}$, and therefore helps prevent adjusted initial samples from becoming OOD. To empirically validate this, we conducted an experiment comparing 5000 unadjusted and 5000 adjusted initial noise samples. We measured the Jensen–Shannon divergence (JSD) between the two resulting distributions, varying the weight decay parameter $w$. The results, shown below, indicate that as $w$ increases, JSD decreases. This suggests that stronger weight decay encourages the adjusted samples to remain closer to the original (in-distribution) noise distribution.
>
> |     | w=0.01 | w=0.05 | w=0.1  | w=0.2 |
> |-----|--------|--------|--------|-------|
> | JSD | 0.0665 | 0.0663 | 0.0638 | 0.046 |
>
> Finally, to examine the effect of weight decay on mitigation performance, we varied the weight decay parameter $w$ as described above and measured the corresponding performance. The table below presents the results of this experiment.
>
> |             | w=0.01   | w=0.05   | w=0.1    | w=0.2    |
> |-------------|----------|----------|----------|----------|
> | SSCD        | 0.2265   | 0.2265   | 0.2265   | 0.2275   |
> | CLIP        | 0.2647   | 0.2647   | 0.2647   | 0.2657   |
> | FID         | 111.7207 | 111.7221 | 111.7221 | 112.0909 |
> | LPIPS       | 0.7615   | 0.7615   | 0.7615   | 0.7610   |
> | ImageReward | -0.3522  | -0.3522  | -0.3522  | -0.3501  |
>
> Across the experimental settings we used, the performance remained relatively consistent regardless of the value of $w$. We believe this is because the default value used in our experiments, $w=0.01$, does not cause the adjusted initial noise to significantly deviate from the original distribution $\mathcal{N}(0, I)$. Indeed, as shown in the JSD results table above, the JSD at $w=0.01$ is 0.0665, indicating minimal distributional shift.
>
> > **Q1-3**: A reason why this type of regularization is not applied.
>
> We did not explicitly incorporate a regularization term to ensure that the initial noise $x_T$ remains within the high-likelihood region of the Gaussian distribution. However, as mentioned in our response to the previous question, the experimental setting used for performance evaluation in this paper (AdamW with $w = 0.01$) does not cause the adjusted initial noise to deviate significantly from the original Gaussian distribution. In fact, it results in a low JSD value.
>
> Moreover, even when we increased the value of $w$—thereby encouraging the adjusted initial noise $x_T$ to stay closer to the high-likelihood region of the Gaussian distribution—no significant performance improvement was observed. These findings suggest that the adjusted initial samples are unlikely to fall into an out-of-distribution (OOD) state that might trigger "reward hacking" behavior.
>
> That said, as you pointed out, explicitly regularizing the adjusted initial noise $x_T$ to remain in-distribution—for example, by adding the regularization term $-\log{p(||x_T||)}$—could contribute to safer deployment of diffusion models. Since
>
> $-\log{p(||x_T||)}= c - { (d-1)\log{||x_T||} + {{||x_T||^2 \over 2}}}$,
>
> and weight decay already minimizes the ${{||x_T||^2 \over 2}}$ term, it should be straightforward to incorporate the full regularizer by augmenting the existing metric $||\tilde{\epsilon}_\theta(x_T, T, y)||_2$ with the additional term $- { (d-1)\log{||x_T||}}$.
>
> > **Q2**: Decomposing evaluation into quality and diversity.
>
> As suggested, we evaluated performance using LPIPS and ImageReward. For both metrics, higher values are better. Despite similar CLIP and ImageReward scores, methods like Adding 4 Random Tokens, Prompt Embedding Adjustment, and Per-sample show notable differences in SSCD and LPIPS, underscoring the strength of our approach.
>
> |                                                 | SSCD   | CLIP   | LPIPS  | ImageReward |
> |-------------------------------------------------|--------|--------|--------|-------------|
> | No Mitigation                                   | 0.5439 | 0.3012 | 0.4835 | -0.014      |
> | Adding 1 Random Token                           | 0.4794 | 0.2903 | 0.5471 | -0.1013     |
> | Adding 4 Random Tokens                          | 0.3402 | 0.2608 | 0.6671 | -0.3532     |
> | Prompt Embedding Adjustment (l_target=3)        | 0.1325 | 0.2372 | 0.7522 | -0.6466     |
> | Prompt Embedding Adjustment (l_target=5)        | 0.2527 | 0.2644 | 0.7003 | -0.3338     |
> | Cross-attention Scaling                         | 0.2779 | 0.2661 | 0.7194 | -0.2115     |
> | Opposite Guidance with Dynamic Transition Point | 0.2396 | 0.2409 | 0.7502 | -0.6639     |
> | Ours (Batch-wise)                               | 0.2642 | 0.2603 | 0.7384 | -0.4527     |
> | Ours (Per-sample)                               | 0.2265 | 0.2647 | 0.7615 | -0.3522     |
>
> > **Q3**: Synergy with other methods.
>
> As shown in [3], $||\tilde{\epsilon}_\theta(x_T, T, y)||_2$ is typically higher for memorized prompts. Our method reduces this value by adjusting the initial noise $x_T$ for such cases.
>
> This design is compatible with other noise optimization techniques aimed at enhancing visual quality or prompt alignment, as long as they do not increase $||\tilde{\epsilon}_\theta(x_T, T, y)||_2$. Integration can be done by simply adding our term to existing loss functions.
>
> > **Q4**: Generality of the Framework.
>
> If an analogous attraction basin could be defined based on unconditional noise prediction $\epsilon_\theta(x_t, t)$ and large $||\epsilon_\theta(x_t, t)||_2$ values similarly indicate strong steering, our method could be adapted accordingly.
>
> [1] Samuel et al. “Norm-guided latent space exploration for text-to-image generation”. NeurIPS 2023.
>
> [2] Ben-Hanmu et al. "D-Flow: Differentiating through Flows for Controlled Generation". ICML 2024.
>
> [3] Jain et al. "Classifier-Free Guidance inside the Attraction Basin May Cause Memorization". CVPR 2025.

---

> ### Comment · Reviewer_JH8S · 2025-08-06
>
> Thank you for the detailed rebuttal. I have some concerns about the optimization methodology that I'd like to clarify.
>
> **Missing Critical Regularization Component:**
>
> Your rebuttal reveals for the first time that weight decay in the AdamW optimizer is intended to serve as an _implicit regularizer_ to keep the optimized noise latent $z_0$ in-distribution. You correctly argue that this involves minimizing $||z_0||^2$. This is not a minor implementation detail; it is the theoretical bedrock of your optimization strategy as can be seen by the SGD numbers.
>
> You mention that this regularization approximates the likelihood of the chi-squared distribution, but there's an important connection you don't explicitly state: minimizing $||z_0||^2$ is equivalent to maximizing likelihood under the Gaussian prior as for $z_0 \sim \mathcal{N}(0, I)$, the log-likelihood is: $\log p(z_0) = -\frac{d}{2} \log(2\pi) - \frac{||z_0||^2}{2}$.
>
> The regularization through weight decay is not a minor implementation detail but a fundamental component that governs the theoretical soundness of your entire approach. Without this regularization, your method risks producing out-of-distribution noise samples that could lead to unpredictable generation behavior. This component is essential for understanding why your method works reliably and safely, yet readers of your paper would have no knowledge of this critical aspect. The omission of such a core theoretical element from the original submission indicates that the methodological presentation could benefit from more comprehensive documentation of key design choices.
>
> **Adam Numbers Discrepcancy:** Your table shows AdamW and Adam achieve identical results across all metrics (0.2265 SSCD, 0.2647 CLIP, 111.7207 FID). This would suggest that actually the weight decay has no impact? Could you please clarify this discrepancy?
>
> **Recommendation:** This regularization framework should be prominently featured in any revision as it's essential for understanding the method's theoretical foundations. I believe the framing should include the goal of keeping $z_0$ in-distribution and then either maximize likelihood under the initial Gaussian, or the Chi-squared, and use the corresponding regularization term. Ideally including an ablation of which of the impact of these terms, and that this is crucial to the method working as suggested by the SGD numbers.
>
>
> **LPIPS computation**: What exactly are you computing LPIPS between here?

---

> > ### Author Response · Authors · 2025-08-07
> >
> > We sincerely appreciate the reviewer’s constructive feedback. Please find our responses to the points raised below.
> >
> > - **Missing Critical Regularization Component**
> >
> >     Thank you for kindly highlighting the connection between minimizing $||x_T||^2$ and maximizing the likelihood under the Gaussian prior. We fully agree with your point regarding the importance of regularization through weight decay. While this aspect was not included in our original submission, we will ensure that it is incorporated into the revised version, as you recommended in your comments (**Recommendation**). We are sincerely grateful for your clear and thoughtful clarification.
> >
> > - **Adam Numbers Discrepcancy**
> >
> >     To begin, we conducted an experiment to assess the extent of distribution shift when adjusting the initial noise using Adam without weight decay. Specifically, we measured the Jensen–Shannon Divergence (JSD) between two distributions: one formed by 5000 unadjusted samples and the other by 5000 adjusted samples (using Adam). This experimental setup is consistent with the one described in our response to Question 1-2 in the first rebuttal which addresses the effect of weight decay. To provide a comprehensive view of how distribution shift varies with different weight decay settings, we present the JSD values—both those obtained using various weight decay values (as presented earlier) and those obtained using Adam without weight decay—in the table below.
> >
> >     |  | Adam | AdamW (w=0.01) | AdamW (w=0.05) | AdamW (w=0.1) | AdamW (w=0.2) |
> >     | --- | --- | --- | --- | --- | --- |
> >     | JSD | 0.0669 | 0.0665 | 0.0663 | 0.0638 | 0.046 |
> >
> >     In the table above, $w$ denotes the weight decay value. As the results indicate, higher weight decay values lead to lower JSD scores, which is consistent with the theoretical understanding of how weight decay helps maintain in-distribution behavior. To further explore the impact of these differences on the generated images, we conducted a performance evaluation using the same experimental setup. The results are provided below.
> >
> >     |  | Adam | AdamW (w=0.01) | AdamW (w=0.05) | AdamW (w=0.1) | AdamW (w=0.2) |
> >     | --- | --- | --- | --- | --- | --- |
> >     | SSCD | 0.2265 | 0.2265 | 0.2265 | 0.2265 | 0.2275 |
> >     | CLIP | 0.2647 | 0.2647 | 0.2647 | 0.2647 | 0.2657 |
> >     | FID | 111.7207 | 111.7207 | 111.7221 | 111.7221 | 112.0909 |
> >     | LPIPS | 0.7615 | 0.7615 | 0.7615 | 0.7615 | 0.7610 |
> >     | ImageReward | -0.3522 | -0.3522 | -0.3522 | -0.3522 | -0.3501 |
> >
> >     As shown in the results above, the performance differences across the metrics—SSCD, CLIP, FID, LPIPS, and ImageReward—were not substantial, even when comparing Adam to AdamW with a weight decay of up to 0.2. We believe this is likely because Adam, although it does not incorporate weight decay, does not move the adjusted initial noise significantly away from the original distribution $\mathcal{N}(0, I)$. Supporting this, the JSD value for Adam remains quite low at 0.0669, as reported in the table.
> >
> >     In summary, although weight decay does influence the distribution of the adjusted initial noise, we did not observe notable differences in the generated images—in terms of memorization, prompt alignment, diversity, and quality—even when weight decay was not applied (i.e., when using Adam). We suspect this is because the adjusted initial noise produced by Adam remains relatively close to the original distribution $\mathcal{N}(0, I)$. This is further supported by our JSD analysis, which shows a consistently low value of 0.0669 for Adam.
> >
> > - **LPIPS computation**
> >
> >     We assessed image diversity using the LPIPS metric, following the procedure described below.
> >
> >     For each of the 500 memorized prompts, we generated 10 images and calculated the LPIPS scores among the images produced for the same prompt. The final diversity score was obtained by averaging these LPIPS values across all 500 prompts.

---

> > > ### Comment · Reviewer_JH8S · 2025-08-07
> > >
> > > Thank for the added analysis of the weight decay and Adam! It is interesting that it doesn't seem to have a big influence on the final results. One potential interpretation here could be that the objective being optimized is not easily hackable, in the sense that an out of distribution noise $z$ could generate high rewards even though the images have e.g. weird artifacts. This could also be investigated further. I'm curious why the SGD numbers are so different then, I assume this is then based on the optimization hyperparameters, e.g. the learning rate tuning? Potentially, the SGD optimization could also be made more stable with an added regularization term?
> > >
> > > In general as mentioned earlier, a thorough discussion of the effect of regularization, why it might be needed or not here, would add a lot of clarity.
> > >
> > > **Regarding LPIPS**: The way you describe it, I would interpret a higher LPIPS score as more similar images to each other and thus, less diverse generations. You mention "higher values are better", am I misunderstanding anything here?

---

> > > > ### Author Response · Authors · 2025-08-08
> > > >
> > > > Thank you for your continued interest in our paper and for actively participating in the discussion. Our responses to the two questions you raised are provided below.
> > > >
> > > > - **SGD Numbers Discrepancy**
> > > >
> > > >     In our response to Question 1-2 of the earlier rebuttal (regarding the choice of AdamW), the performance evaluation table we provided was based on experiments using a learning rate of 0.01 for all optimizers. In those experiments, the performance gap between AdamW and SGD occurred since SGD (with a learning rate of 0.01) was not effective in reducing $||\tilde{\epsilon}_\theta(x_T, T, y)||_2$.
> > > >
> > > >     As you suggested, after adjusting the learning rate, we found that SGD can also effectively minimize $||\tilde{\epsilon}_\theta(x_T, T, y)||_2$ and generate non-memorized images.
> > > >
> > > >     Due to the limited time remaining in the discussion period, we are unable to determine the optimal learning rate for SGD and present the corresponding results at this stage.
> > > >
> > > >     However, we believe that, with an appropriately tuned learning rate that effectively reduces $||\tilde{\epsilon}_\theta(x_T, T, y)||_2$,
> > > >
> > > >     SGD can also achieve strong mitigation performance. Furthermore, incorporating a regularization term that explicitly encourages the reduction of $||\tilde{\epsilon}_\theta(x_T, T, y)||_2$ could further enhance the stability and effectiveness of SGD optimization.
> > > >
> > > > - **Regarding LPIPS**
> > > >
> > > >     As described in Zhang et al. [1], the LPIPS score we employ can be described, in simple terms, as the $\ell_2$ distance between the feature embeddings of two given images. A smaller $\ell_2$ distance indicates that the two images are more similar, whereas a larger distance indicates that they are more different. Consequently, a higher LPIPS score means the compared images are more dissimilar, which in turn reflects greater diversity.
> > > >
> > > >     In general, Stable Diffusion models tend to generate memorized images when given memorized prompts, which can reduce the diversity of the generated outputs. As noted in Section 1, this occurs because memorization leads to reproducing training data, thereby limiting both the diversity and the model’s capacity to synthesize truly novel content. However, when a memorization mitigation method is applied, which allows the generation of non-memorized images for memorized prompts, the diversity of the generated outputs is expected to increase.
> > > >
> > > >     Consistent with this, the table presented in our earlier response to Question 2 of the rebuttal (Decomposing evaluation into quality and diversity) shows that all mitigation methods yield higher LPIPS scores compared to *No Mitigation*. This suggests that mitigating memorization contributes to generating more diverse images.
> > > >
> > > >
> > > > [1] Zhang et al. “The Unreasonable Effectiveness of Deep Features as a Perceptual Metric”. CVPR 2018.

---

> > > > > ### Comment · Reviewer_JH8S · 2025-08-08
> > > > >
> > > > > Thank you for the thorough responses and additional experiments. Your clarifications on the weight decay regularization and optimizer comparisons were helpful in understanding the method's theoretical foundations.
> > > > >
> > > > > Thank you for the clarification on LPIPS computation. You're absolutely right that higher LPIPS scores indicate greater diversity between images generated from the same prompt, which is indeed desirable for memorization mitigation. I appreciate the correction on my interpretation. Regarding the SGD performance differences, your explanation that this was due to learning rate tuning (0.01 for all optimizers) makes sense.
> > > > >
> > > > > I appreciate your commitment to incorporating the regularization discussion into the revised version, as this will help readers understand the method's design principles. The connection between weight decay and Gaussian likelihood maximization that emerged in our discussion would be valuable to include in the methodology section.
> > > > >
> > > > > The additional LPIPS and ImageReward evaluations strengthen the empirical analysis, and your explanations about the optimization details provide useful insights into the method's behavior. I'm keeping my score of 4 and look forward to seeing the revised version with the enhanced theoretical presentation you've outlined.

---

> > > > > > ### Author Response · Authors · 2025-08-09
> > > > > >
> > > > > > We are glad to hear that our responses have addressed all of your concerns. We will incorporate the points discussed into the revised version. We sincerely appreciate your active engagement throughout the discussion and your contributions toward improving the quality of our paper. We are also grateful for your consistently positive evaluation from the beginning to the end of the review process.

---

### Decision · Program_Chairs · 2025-09-17

**Decision:**

Accept (poster)

**Comment:**

This work aims to reduce memorization in diffusion-based text-to-image generation by proposing a method that adjusts the initial noise (either at a batch-level or sample-level) which encourages the model to escape the attraction basin. The method is novel, has solid theoretical groundings and experimental results are satisfactory. Discourse with reviewers have mostly addressed all concerns. Although the relevance of the work may be diminishing given the latest image-generation models don't have a severe memorization issue, I am willing to recommend acceptance due to the algorithmic contributions of this paper.